# Low CO$_2$ levels of the entire Pleistocene epoch

Jiawei Da [1], Yi Ge Zhang [2], Gen Li[3], Xianqiang Meng[1] & Junfeng Ji[1]*

Quantifying ancient atmospheric $pCO_2$ provides valuable insights into the interplay between greenhouse gases and global climate. Beyond the 800-ky history uncovered by ice cores, discrepancies in both the trend and magnitude of $pCO_2$ changes remain among different proxy-derived results. The traditional paleosol $pCO_2$ paleobarometer suffers from largely unconstrained soil-respired $CO_2$ concentration ($S(z)$). Using finely disseminated carbonates precipitated in paleosols from the Chinese Loess Plateau, here we identified that their $S(z)$ can be quantitatively constrained by soil magnetic susceptibility. Based on this approach, we reconstructed $pCO_2$ during 2.6–0.9 Ma, which documents overall low $pCO_2$ levels (<300 ppm) comparable with ice core records, indicating that the Earth system has operated under late Pleistocene $pCO_2$ levels for an extended period. The $pCO_2$ levels do not show statistically significant differences across the mid-Pleistocene Transition (*ca.* 1.2–0.8 Ma), suggesting that $CO_2$ is probably not the driver of this important climate change event.

[1] Key Laboratory of Surficial Geochemistry, Ministry of Education, School of Earth Sciences and Engineering, Nanjing University, 210023 Nanjing, China. [2] Department of Oceanography, Texas A&M University, College Station, TX 77843, USA. [3] Division of Geological and Planetary Sciences, California Institute of Technology, Pasadena, CA 91125, USA. *email: jijunfeng@nju.edu.cn

Over the last 3 million years (Ma), the Earth has cooled substantially and transitioned from a single-pole glaciation to a stage with major glaciations on both poles. In the past 1 Ma, the period of glacial cycles switched from ~41 thousand years (ky) to ~100 ky (mid-Pleistocene Transition, MPT), with the magnitude of the climate change associated with the glacial cycles significantly enhanced[1,2]. What has ultimately driven these far-reaching climate changes is unclear. Atmospheric $CO_2$, the most important non-condensing greenhouse gas, plays an important role in global climate changes. However, although an overall decreasing trend has been documented, the available early Pleistocene (2.6–0.8 Ma) $pCO_2$ reconstructions, mostly from the marine realm, differ for both trends and absolute values[3].

As a terrestrial archive, the pedogenic carbonates found in paleosols have been widely applied for $pCO_2$ reconstruction[4], especially for the Paleozoic and Mesozoic eras when other proxies are often not available[5]. In soils, the rate of carbonate crystallization is much slower than gas exchange, thus pedogenic carbonates precipitated in the vadose zone are thought to reach carbon isotopic equilibrium with $CO_2$ in soil pore space[6]. This soil-pore $CO_2$ represents a mixture of two end members with distinct $\delta^{13}C$ signatures: atmospheric $CO_2$ and soil-respired $CO_2$. Built on this isotopic end-member mixing principal and gas diffusion dynamics, previous studies have derived a complete solution to estimate ancient atmospheric $pCO_2$ from soil-associated parameters (i.e. the paleosol $CO_2$ paleobarometer[7]). Specifically, atmospheric $pCO_2$ levels can be expressed as

$$pCO_2 = S(z)\frac{\delta^{13}C_s - 1.0044\delta^{13}C_r - 4.4}{\delta^{13}C_a - \delta^{13}C_s} \quad (1)$$

where $\delta^{13}C_s$, $\delta^{13}C_r$, and $\delta^{13}C_a$ refer to the carbon isotopic compositions of total soil $CO_2$, soil-respired $CO_2$, and atmospheric $CO_2$, respectively, and $S(z)$ is the soil-respired $CO_2$ concentration at soil depth $z$ during the time of carbonate precipitation. Generally, the soil $\delta^{13}C_s$ is calculated from $\delta^{13}C_c$ (the carbon isotopic composition of pedogenic carbonate) using a temperature-dependent fractionation factor[8]; the soil-respired $\delta^{13}C_r$ is approximated by $\delta^{13}C$ value of coeval soil organic matter (SOM) preserved in the same paleosol strata[9]; and the carbon isotopic composition of atmospheric $CO_2$ ($\delta^{13}C_a$) can be estimated from marine carbonate records[10]. This approach has been extensively applied to pedogenic carbonates worldwide[11] to reconstruct $pCO_2$ over the geological past. Approaches to determine the carbon isotopic terms ($\delta^{13}C_s$, $\delta^{13}C_r$, and $\delta^{13}C_a$) have been well established. However, significant uncertainties still remain in this paleosol $CO_2$ barometer, which is mainly sourced from the $S(z)$ term that describes soil respiration[12].

$S(z)$ has been difficult to constrain. Early studies[11] commonly treated $S(z)$ as a constant of 5000 ppm based on limited modern observations[13]. However, accumulating evidence over the last decade has shown that $S(z)$ is highly variable both spatially and temporally[14,15]. Studies of Holocene calcareous soils demonstrated that pedogenic carbonate precipitates under warm, dry climatic conditions, when soil productivity, and $S(z)$ are much lower than previously assumed[14]. Besides, among different soil types and soil moisture regimes, $S(z)$ can vary by up to two orders of magnitudes[16]. The extensive variability of $S(z)$ across different climatic conditions and soil types leads to large uncertainties of the paleosol-based method, especially for the late Cenozoic when $pCO_2$ is low with small variations[17]. Instead of treating $S(z)$ as a constant, recent works took into account of the variability of $S(z)$ either by establishing empirical models equating $S(z)$ with other climatic parameters such as mean annual precipitation (MAP)[18], or by defining soil order-specific ranges[15]. Although a major step forward, both approaches have caveats. Estimates of $S(z)$ using

MAP proxies such as CIA-K (chemical index of alteration without potassium)[18] or DTC (depth to carbonate nodular horizon)[19], are empirically developed based on modern soil systems, but have not been validated for applications in paleosols[20]. $S(z)$ values of different soil types, for example, the aridisols, still vary from 500 to 2500 ppm[15], which would contribute to a five-fold spread in estimated $pCO_2$. Importantly, the calculated $pCO_2$ scales proportionately with $S(z)$ (Eq. (1)), making it a vital factor for the success of paleosol-derived $CO_2$ estimates.

In this study, we seek to improve the paleosol $CO_2$ barometer by providing more robust constraints on $S(z)$. To achieve this, we focus on the finely disseminated carbonates (FDC) in bulk paleosols on the Chinese Loess Plateau (CLP). Field observations and scanning electronic microscopy (SEM) imaging together demonstrate a shallow formation depth of FDC (soil Bt/Bw horizon)[21]. Further, a suite of geochemical analyses are used to test the pedogenic origin of these carbonates, as well as the potential of diagenetic alterations. Rather than assuming a constant $S(z)$ for all samples, we explore the possibility of sample-specific $S(z)$ values by employing soil magnetic susceptibility (MS) as a proxy for $S(z)$. With this refined method and new data, we intend to provide a better assessment of the interglacial $pCO_2$ for the early Pleistocene epoch from a terrestrial perspective.

## Results

**Geological setting and sampling.** The accumulation of eolian material on the CLP initiated during the early Miocene[22]. In the Quaternary loess–paleosol sequence, paleosol units formed during interglacial episodes were characterized by slower eolian dust deposition, strengthened summer monsoonal rainfall, and enhanced pedogenesis relative to the glacials[23]. Large seasonal changes in precipitation and evaporation contribute to the wide occurrence of pedogenic carbonates across the CLP. Unlike the more humid southeastern CLP (e.g. typical sites including Lantian, Baoji), where extensive leaching completely decalcified the paleosols, or the more arid northwestern CLP (e.g. Huanxian, Jingyuan), where large amount of detrital carbonates remained due to insufficient rainfall[24], our sampling location, the Luochuan section (35.76°N, 109.42°E) located in the central CLP (Supplementary Fig. 1), is characterized by the appropriate level of pedogenesis. The detrital carbonates are mostly absent in paleosol samples, whereas authigenic carbonates are still preserved (Supplementary Fig. 2). Before sampling, we trenched the soil profile (>1 m deep) to avoid contamination of regolith, and monitored the MS values of bulk paleosols, which were compared to published data[25]. Each sample was collected within a 10 cm interval. Paleosol samples used in this study mainly occur within the soil Bt/Bw horizons, characterized by subangular blocky structure with clay coatings attached on structural faces, and a 5–7.5 YR Munsell color.

**Finely disseminated carbonates.** In the field, the carbonates in our targeted paleosol samples have identifiable pedofeatures including finely disseminated cements filling the soil matrix, and carbonate hypocoatings (i.e. pseudomycelia) lining pores and tubules, which can be categorized as Stage 1 pedogenic carbonate[26]. Therefore, we term this carbonate fraction as FDC. SEM imaging further reveals that our sampled FDC are mainly composed of 5–10 μm-long needle fiber calcite (NFC), and clustered nanoscale calcite nanofibers (Fig. 1a–d). The SEM results are consistent with previous findings that NFC is commonly associated with smaller carbonate fibers such as nanofibers[27], indicating similar formation conditions and a pedogenic origin. Importantly, NFC mostly appears between the humic horizon (O horizon) and subsurface C zone and vanishes with depth[28]. The

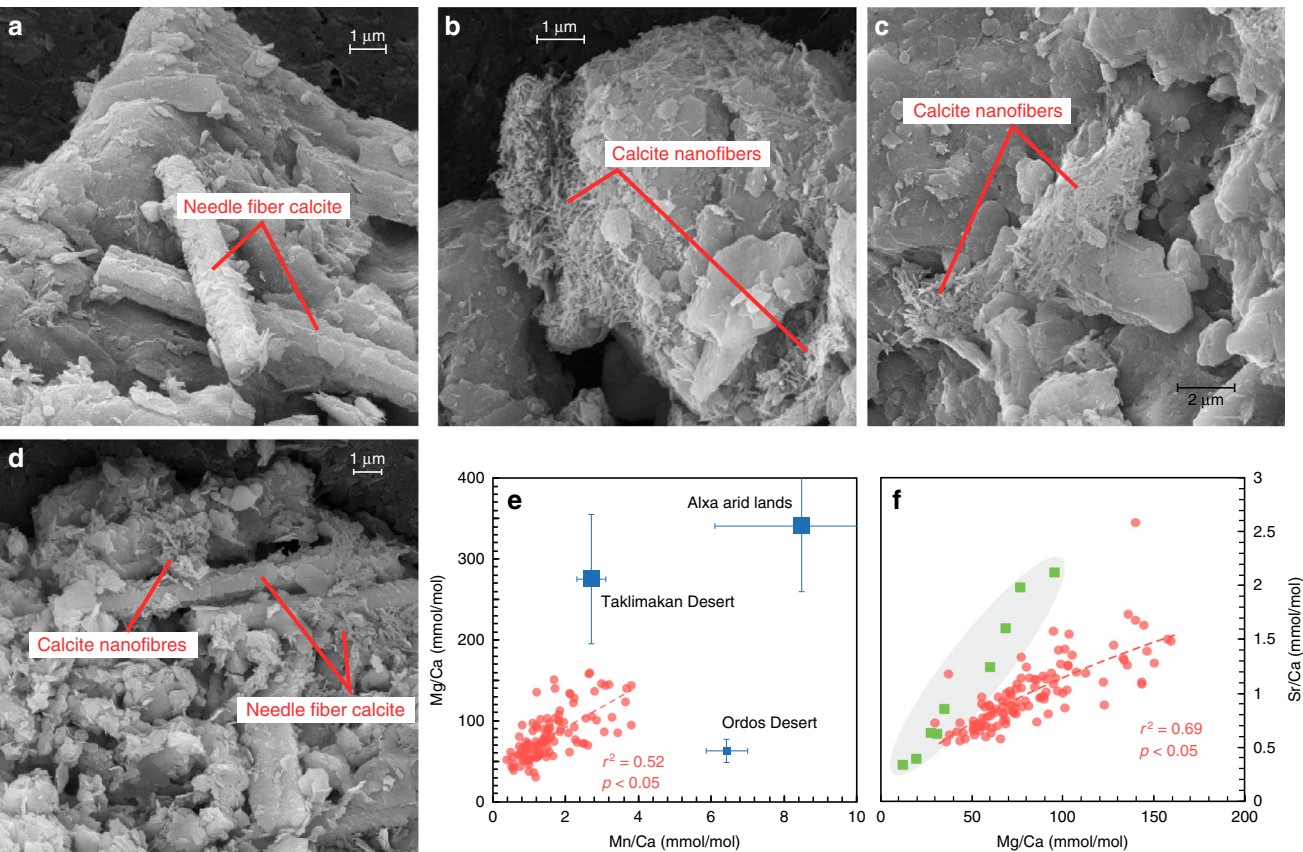

**Fig. 1** Micromorphology and geochemistry of pedogenic carbonates. **a** Needle fiber calcites (NFC) in the inner channel of hypocoatings. **b** and **c** Randomly organized mesh of calcite nanofibers. **d** Sparsely distributed nanofibres lying on the surfaces of NFC. **e** Cross plots of Mn/Ca and Mg/Ca ratios of finely disseminated carbonate (FDC) samples in this study (red dots), compared with detrital carbonates from potential source regions (blue squares)[32]. Error bars represent standard deviations from multiple samples ($n > 3$). **f** Cross plots of Sr/Ca and Mg/Ca ratios of FDC samples in this study, compared with those of microcodium (green squares) in Holocene soils on the Chinese Loess Plateau[33]

location of NFC matches our field observations, which indicate the occurrence of FDC within the relatively shallow soil Bt/Bw horizon in comparison to the nodule-bearing Bk horizon. The origin of NFC is tied to both organic (fungal biomineralization) and inorganic activities (physicochemical precipitation)[29,30]. Whichever the case, it appears that the stable isotopic compositions ($\delta^{13}C$ and $\delta^{18}O$) of NFC are indistinguishable from coeval calcite cements (rhombohedral calcite crystals)[31], making bulk measurements of FDC well representative of all encapsulated calcite phases.

Incomplete dissolution of the detrital carbonate components (calcite and dolomite) from parent material of the loess (e.g. eolian dust) would bias the $\delta^{13}C$ of the bulk soil carbonates. However, they appear to only impose very minor influence on our samples because we selected samples without the presence of any dolomite for subsequent analyses, as measured by the Fourier transform infrared spectrophotometry that is highly sensitive to carbonates[24]. Dolomite has a slower dissolution rate than calcite, and the final disappearance of detrital dolomite ensures the complete dissolution of detrital carbonates[24]. Further, we also measured trace elements of the FDC fractions in our paleosol samples. Previous work has shown that the concentrations of trace elements (e.g. Mn, Mg, and Sr) in pedogenic carbonates from the CLP are considerably lower than those of detrital carbonates derived from marine carbonate strata in source regions[32], and are highly correlated with rainfall intensity[33]. In this study, we identified that the Sr/Ca and Mg/Ca ratios of the FDC fractions range from 0.72 to 1.48 mmol/mol (mean = 0.98 ± 0.31 mmol/mol) and 36–140 mmol/mol (mean = 81 ± 30 mmol/mol),

respectively, both of which are generally consistent with those of microcodium (typical authigenic carbonate in paleosols) in Holocene soils on the CLP (Sr/Ca = 1.08 ± 0.68 mmol/mol, Mg/Ca = 48 ± 29 mmol/mol) (Fig. 1f)[33]. The Mn/Ca and Mg/Ca ratios of the FDC fractions are significantly lower than the carbonate samples from potential source regions of the loess— the deserts in Northern China (Fig. 1e)[32], providing additional evidence for the pedogenic origin of our bulk paleosols carbonates.

Diagenetic alterations (i.e. the dissolution and reprecipitation of fine-grained carbonates) could also affect $\delta^{13}C$ and other geochemical signatures. However, we expect this process to have minimal influence on our carbonate samples for two reasons. Firstly, the long-term trend of $\delta^{13}C$ derived from the FDC fractions of our paleosol samples are almost identical with that determined from $\delta^{13}C$ of coeval calcite nodules throughout the Pleistocene, although the nodule record has a much lower resolution (Supplementary Fig. 3). Calcite nodules are usually considered to be buffered against diagenesis[11,18]. Notably, the Pliocene paleosol formations in the CLP also show indistinguishable $\delta^{13}C$ and $\delta^{18}O$ values from mm-scale micrites to massive carbonate horizons[34], suggesting that the FDC could potentially be used to reconstruct $pCO_2$ beyond the Pleistocene. Secondly, numerous studies of paleosol carbonates from the CLP suggest that their geochemical signals are pristine and record near-surface conditions. For instance, trace metal compositions of fine-grained pedogenic carbonates from loess–paleosol sections in the CLP appear to preserve local precipitation signals[33] and show high-frequency variations on orbital timescales[35].

Based on field observations, micromorphology, and geochemical characteristics, we conclude that the FDC fractions used in this study are of a pedogenic origin with minimal diagenetic alteration, thus suitable for paleosol-based $p\mathrm{CO}_2$ reconstructions.

**Resolving $S(z)$.** Rearranging Eq. (1) and we obtain

$$S(z) = p\mathrm{CO}_2 \times \frac{\delta^{13}\mathrm{C}_a - \delta^{13}\mathrm{C}_s}{\delta^{13}\mathrm{C}_s - 1.0044\delta^{13}\mathrm{C}_r - 4.4} \quad (2)$$

which provides a mathematical solution for $S(z)$. Carbonate and SOM were measured for stable C isotopes to obtain $\delta^{13}\mathrm{C}_s$, and $\delta^{13}\mathrm{C}_r$ (see the "Methods" section). $\delta^{13}\mathrm{C}_a$ was obtained from marine carbonate $\delta^{13}\mathrm{C}$[10]. Atmospheric $p\mathrm{CO}_2$ over the last 800 ky were derived from the ice core $p\mathrm{CO}_2$ record[36]. Together they were used to compute $S(z)$ of the Luochuan section over the last 800 ky. To estimate the uncertainties of the back-calculated $S(z)$, we adopted Monte Carlo random sampling simulations to propagate errors from all input parameters (Supplementary Note 1). In brief, for each sample, we randomly sampled each parameter in Eq. (2) within its error range for 10,000 times and calculated a corresponding $S(z)$ population. We report $S(z)$ as the median values of the results from the Monte Carlo simulations, and define the associated uncertainties using the 16th and 84th percentiles of the simulated $S(z)$ population (corresponding to the median value and the $\pm 1\sigma$ error range in a standard Gaussian distribution).

When pedogenic carbonate precipitates deep in the soil profile where soil-respired $\mathrm{CO}_2$ dominates over atmospheric $\mathrm{CO}_2$, back-calculated $S(z)$ are inherently associated with sizable errors[7]. We therefore adopted the concept of an $R$ factor, defined as the ratio between $[\mathrm{CO}_2]_{\mathrm{atm}}$ and $S(z)$ ($R = (\delta^{13}\mathrm{C}_s - 1.0044 \times \delta^{13}\mathrm{C}_r - 4.4)/(\delta^{13}\mathrm{C}_a - \delta^{13}\mathrm{C}_s)$), as a screening criterion[7]. When atmospheric $\mathrm{CO}_2$ constitute a minor portion of soil $\mathrm{CO}_2$ ($R < 0.3$ according to ref. [7]), the samples are thought to be dominated by soil-respired $\mathrm{CO}_2$ thus discarded for subsequent analyses. Twenty-two paleosol samples passed this criterion, representing 78% of all the samples analyzed for $S(z)$. The calculated median $S(z)$ levels over the last 800 ky show variations from 356 to 815 ppm, with uncertainties on average of $+96/-75$ ppm (Fig. 2a).

**MS as a proxy for $S(z)$.** Soil-respired $\mathrm{CO}_2$ comes from two major sources—the respiration of autotrophs, such as plant roots, and

heterotrophs, such as the soil microbes mediating the oxidation of SOM. Studies of modern soils have shown that $S(z)$ as a measure of soil productivity, is strongly correlated to climate parameters, such as temperature, precipitation, and evapotranspiration[37]. Notably, among carbonate-bearing soils in semi-arid to arid regions, such as the CLP, $S(z)$ is mainly controlled by precipitation[18,37]. To quantitatively constrain $S(z)$, here we explore a potential proxy sensitive to precipitation changes—soil magnetic susceptibility (MS).

The MS of bulk soil samples is predominantly controlled by the presence of ultrafine magnetite[38,39], which is efficiently produced in situ under well-drained soils with alternative wet ($\mathrm{Fe}^{2+}$ produced) and dry ($\mathrm{Fe}^{2+}/\mathrm{Fe}^{3+}$ oxide precipitated) cycles. The ultrafine ferromagnetic grains are either directly produced by magnetotactic bacteria, or formed by inorganic precipitation, mediated by the production of $\mathrm{Fe}^{2+}$ through iron-reducing bacteria[40]. Similarly, modest precipitation increase would enhance microbial activities, the formation of magnetic iron oxides and consequently soil MS, but MS decreases under excessive rainfall (MAP > 1000 mm) as the production of ferrimagnet ceases[41]. MAP > 1000 mm probably never occurred in the Pleistocene history of the central to northern part of the CLP[42]. As a result, MS is regarded as a sensitive paleoclimate proxy of pedogenesis intensity and paleorainfall[40], and has been widely used to address the evolution history of the East Asian summer monsoon[23,43].

Intensified precipitation would facilitate the formation of magnetic iron oxides (i.e. MS), and increase soil productivity and $S(z)$. Because rainfall regulates soil MS and $S(z)$ in a similar manner on the CLP, these two parameters are expected to be correlated. Indeed, when the two parameters from the Luochuan samples covering the last 800 ky were plotted against each other, we observe a statistically significant correlation (Fig. 2a):

$$S(z) = 2.66(\pm 0.44) \times \mathrm{MS} + 114.9(\pm 71.1)(r^2 = 0.64, p < 0.0001) \quad (3)$$

The fitting is done using the least-squares linear regression model, and the uncertainties on the slope and intercept represent $1\sigma$ standard errors. Extrapolating this relationship to the entire Pleistocene will enable sample-specific $S(z)$ using measured MS, and consequently the calculation of $p\mathrm{CO}_2$ from paleosols (MS–$S(z)$ approach). By providing new constraints on $S(z)$, this MS–$S(z)$ approach potentially improves the paleosol method for reconstructing $p\mathrm{CO}_2$.

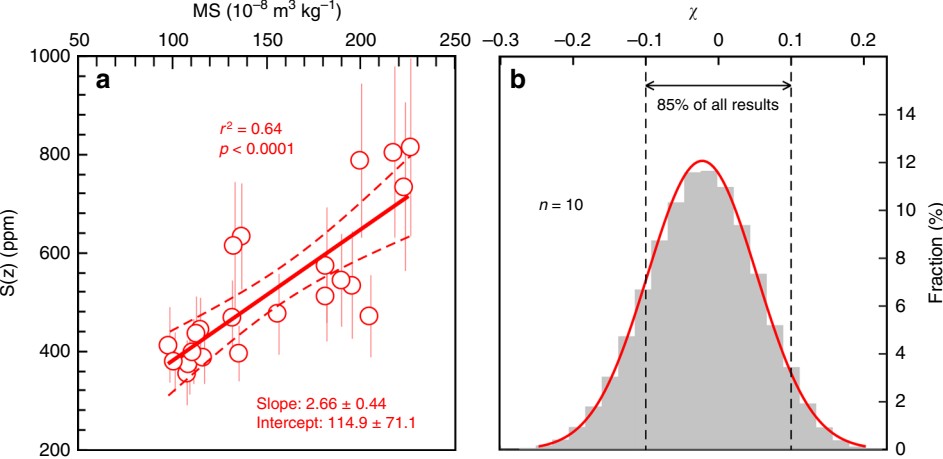

**Fig. 2** The magnetic susceptibility (MS)–$S(z)$ correlation of the Chinese Loess Plateau. **a** Back-calculated $S(z)$ values plotted against MS from Luochuan paleosol samples spanning the last 800 ky. Error bars represent $1\sigma$ errors resulted from Monte Carlo simulations (Supplementary Note 1). The best fitting line, as well as 95% confidence interval are indicated by red solid and dotted lines, respectively. **b** Histogram of the mean relative differences ($\chi$) between calculated and ice core $p\mathrm{CO}_2$, when the sample numbers in the training group ($n$) is set to 10

To validate the MS–S(z) approach, we next recalculated the S(z) of our 800-ky paleosol samples using the MS–S(z) approach, and compare the derived $p$CO₂ to those documented in ice cores. Because the determination of the MS–S(z) relationship relied on the ice core CO₂ record, to avoid circular reasoning, we adopted a resampling technique. Specifically, the 22 samples were divided into two subsets: a training sample group and a test sample group. A MS–S(z) regression was established from the training group, and used to estimate S(z) for the test group, the resulting $p$CO₂ of which were then compared to the ice core data. For a given training sample number $n$, a bootstrap sampling of 1,000,000 times was performed (each time, we obtained $22 - n$ $p$CO₂ values). We also varied the sample numbers in the training group from $n = 10$ to $n = 21$. During each run, the mean relative difference $\chi$ between the calculated $p$CO₂ and those from the ice core data was calculated:

$$\chi = \frac{1}{(22-n)} \times \sum_{i=1}^{22-n} \left(1 - \frac{\text{Calculated } p\text{CO}_2}{\text{Ice core } p\text{CO}_2}\right) \quad (4)$$

The statistical distributions of $\chi$ (Fig. 2b and Supplementary Fig. 4) show that, across different training sample number $n$, $\chi$ is clustered around 0 with >70% of the data points falling within ± 10% difference. This consistent pattern among different $n$ values

strongly supports our MS–S(z) approach for $p$CO₂ reconstruction over the last 800 ky.

**Reconstructing early Pleistocene $p$CO₂.** The refined soil carbonate-$p$CO₂ approach was also applied to the samples from the lower section of the Luochuan profile spanning the early Pleistocene, beyond the available ice core records. Measured MS and stable carbon isotopes for those early Pleistocene samples were used to calculate S(z) and $p$CO₂, assuming that the MS–S(z) relationship remains (Eq. (3)). We calculated the uncertainties on $p$CO₂ by propagating the uncertainties associated with the analyses of all input parameters and the MS–S(z) regression equation using PBUQ, a published MatLab-based program that adopts Monte Carlo random sampling simulations for uncertainty analysis[7]. We report the median values of the calculated $p$CO₂, and define the uncertainties as the 16th and 84th percentiles (as ±1σ), and the 2.5th and 97.5th percentiles (as ±2σ) of $p$CO₂ distributions from Monte Carlo simulations (see the "Methods" section).

The calculated S(z) during ~2.6–0.8 Ma vary between 276 and 644 ppm, with errors (1σ) ranging from 76 to 113 ppm (Fig. 3c). The R values ($p$CO₂/S(z)) are generally higher (>0.3) during the early Pleistocene, suggesting that these samples are suitable for

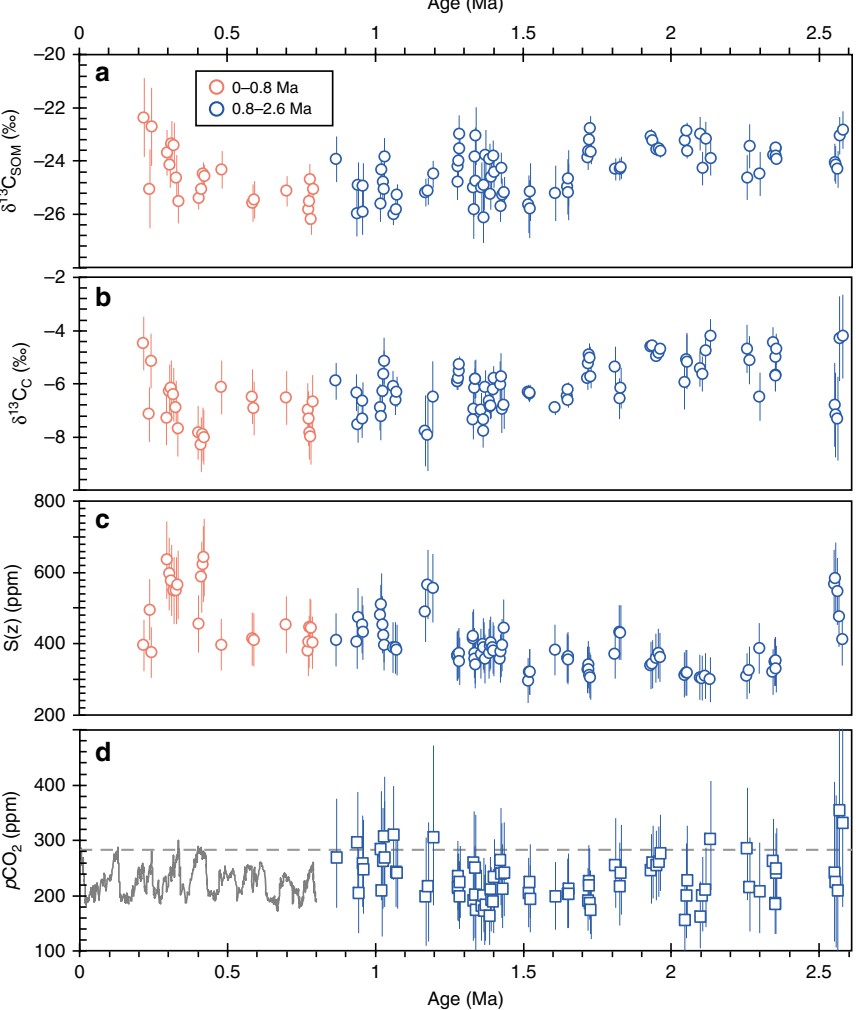

**Fig. 3** Stable isotopes and calculated S(z) and $p$CO₂. **a** δ¹³C values of soil organic matter (δ¹³C$_{SOM}$). **b** δ¹³C values of finely disseminated carbonates (δ¹³C$_c$). **c** S(z) estimates based on the magnetic susceptibility (MS) proxy. **d** Reconstructed early Pleistocene $p$CO₂ (median levels of $p$CO₂ distributions, 2.6–0.8 Ma) and ice core CO₂ record (0.8–0 Ma)[36]. Horizontal gray line shows the pre-industrial $p$CO₂ level (280 ppm). Errors associated with δ¹³C$_c$ and δ¹³C$_{SOM}$ are standard deviations of all measurements within the same paleosol unit ($n > 3$). Error bars related to $p$CO₂ represent the 16th and 84th percentiles (1σ) based on PBUQ Monte Carlo simulations[7], whereas errors associated with S(z) were calculated based on Gaussian error propagation (Supplementary Note 1)

$p$CO$_2$ reconstructions. This newly established interglacial $p$CO$_2$ records show variations from 183 to 292 ppm (averaged median $p$CO$_2$ levels for each interglacial episode) during 2.6–0.9 Ma (Fig. 3d). Except for some data points centered around the Pliocene–Pleistocene boundary (2.6–2.5 Ma) and the MPT showing relatively higher $p$CO$_2$ exceeding 300 ppm, our paleosol-CO$_2$ estimates document overall low early Pleistocene $p$CO$_2$ levels similar to those over the last 800 ky (Fig. 3d). The reconstructed $p$CO$_2$ values have an 1$\sigma$ error range of +92/−82 ppm when averaging the errors for all individual results.

To evaluate the role $p$CO$_2$ played for the MPT, we performed a detailed comparison of MPT–$p$CO$_2$ with those before and after the MPT (i.e. pre-MPT and post-MPT), the durations of which were considered to be 400 ky equivalent to the MPT itself (i.e. 0.8–0.4 Ma for post-MPT and 1.6–1.2 Ma for pre-MPT). Our paleosol-based CO$_2$ reconstructions provide 16 estimates during the MPT and 21 pre-MPT. $p$CO$_2$ estimates during the MPT and pre-MPT were then converted into factor changes in CO$_2$ by normalizing $p$CO$_2$ to the mean post-MPT $p$CO$_2$ from ice cores, since our paleosol–CO$_2$ estimates are indistinguishable from ice core data (Fig. 2b and Supplementary Fig. 4). We calculated the probability density functions (PDFs) and cumulative distribution functions (CDFs) of the factor changes in CO$_2$ among different time periods (i.e. MPT/pre-MPT, MPT/post-MPT, pre-MPT/post-MPT) (Supplementary Note 2). As indicated by both absolute values and the factor changes, $p$CO$_2$ level during the MPT (mean = 269 ± 38 ppm, 1$\sigma$ = +105/−94 ppm) was ~15% higher than that of the post-MPT (mean = 231 ± 37 ppm), and ~20% higher than that of the pre-MPT (mean = 217 ± 25 ppm, 1$\sigma$ = +73/−67 ppm) (Fig. 4a). Results from the PDFs and CDFs also confirm this distribution (Fig. 4b, c).

## Discussion

The foundation of the refined paleosol-based $p$CO$_2$ estimates lies in the premise that soil MS can be used to constrain $S(z)$, with the rationale that on the CLP, both parameters are controlled by monsoonal precipitation. However, soil residence time, or the duration of pedogenesis before they were deeply buried by the newly arrived eolian material, could potentially affect MS, thus hampering the application of our MS–$S(z)$ approach. However, multiple lines of evidence suggest that since soil formation, soil MS quickly reaches a near-steady-state equilibrium status, insensitive to the duration of pedogenesis. For instance, the MS of modern surface soils across the CLP region (30–200)[44] are equivalent to those of the Pleistocene paleosols (60–220) from this study, even though the modern soils have undergone much shorter periods of pedogenesis.

Changes of the accumulation rate of dust supply could also modify soil properties. However, in the central CLP region (e.g. Luochuan in this study), the mean mass accumulation rate of loess remained relatively constant (ca. 5–8 g m$^2$ yr$^{−1}$) without discernable trend during the interglacial cycles over the last 2.6 Ma[45]. As a consequence, we render that the observed correlation between MS and $S(z)$ over the last 800 ky can be applied to early Pleistocene paleosols.

It is noteworthy that interglacial MS levels during the early Pleistocene (99 ± 28) are generally lower than those during the last 800 ky (154 ± 28), and most of the MS values (78 ± 11) of our 1.7–1.5 Ma samples fall below the MS range (99–227) over the last 800 ky. This translates to low $S(z)$, low soil productivity, weak pedogenesis, and perhaps drier climate during the early Pleistocene, which is consistent with our current understanding of the early Pleistocene climate on the CLP[42,46]. For instance, Fe minerals found in the CLP sequences are sensitive to climate changes, as high temperatures and limited seasonal rainfall would favor the

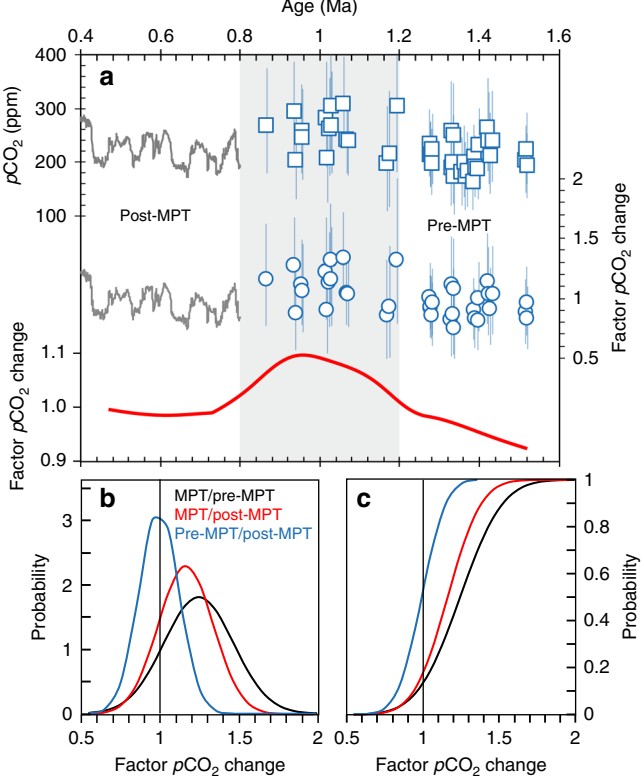

**Fig. 4** $p$CO$_2$ and factor changes of $p$CO$_2$ during 1.6–0.4 Ma. **a** Interglacial paleosol-$p$CO$_2$ estimates from this study (blue squares), ice core $p$CO$_2$ data[36] (gray curves), and their factor changes (blue circles). Error bars represent 1$\sigma$ uncertainties. The three time periods (i.e. pre-mid-Pleistocene transition (MPT), MPT, and post-MPT) are shown separately, with the MPT highlighted by the gray bar. The locally weighted polynomial regression (LOESS) trend line (red curve) is based on 0.3 smoothing. **b** and **c** Probabilities of interglacial CO$_2$ change. The probability density functions (PDFs) and cumulative distribution functions (CDFs) show the probabilities of the factor changes in CO$_2$ among the three time periods (Supplementary Note 2). Ice core data from both glacial and interglacial intervals are presented for a complete view. However, for the calculation of factor changes in CO$_2$, only interglacial CO$_2$ data from ice cores were used to be consistent with our paleosol-based CO$_2$ reconstructions

formation of hematite over goethite. The hematite/goethite ratio of Luochuan section demonstrates a long-term, stepwise decreasing trend from 0.25 to 0.3, since the Pliocene/Pleistocene boundary, to as low as 0.2 towards the late Pleistocene, indicating reduced monsoonal precipitation during the early Pleistocene[47].

Previous estimates of the early Pleistocene $p$CO$_2$ mostly come from the applications of geochemical proxies to marine sediments (e.g. alkenone and boron approaches)[48–52]. Although results from both boron and alkenone-based proxies suggest an overall decreasing trend since the early Pleistocene, discrepancies remain among individual records (Fig. 5). For instance, the composite view from two boron isotope records[48,49] indicates a sudden decline of $p$CO$_2$ at ~2.2 Ma, while the alkenone proxy suggests a gradual decline since ~2.5 Ma. Two alkenone-based results[17,50], however, show early Pleistocene $p$CO$_2$ values higher than 300 ppm, significantly elevated from the late Pleistocene levels recorded by ice cores. Both alkenone and boron proxies rely on certain assumptions, which could be invalidated by local physical, chemical, and biological processes[53]. For example, marine-based $p$CO$_2$ reconstructions assume that an air–sea equilibrium of CO$_2$ is maintained for the entire studied interval, which does not

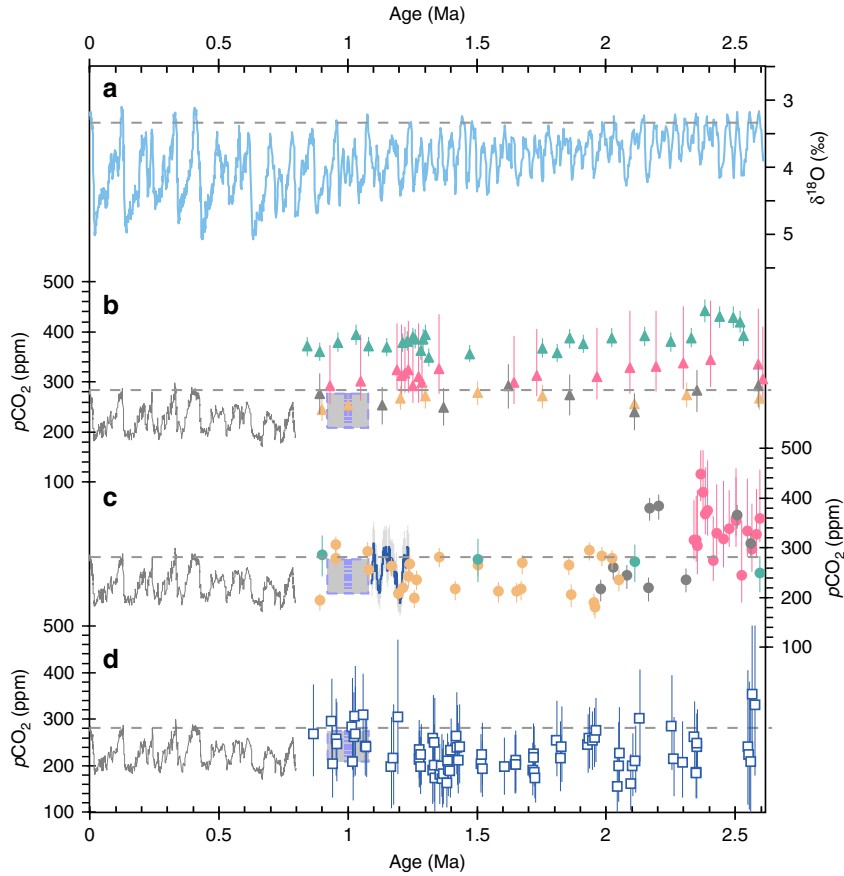

**Fig. 5** $pCO_2$ and benthic $\delta^{18}O$ data since 2.6 Ma. **a** Benthic $\delta^{18}O$ stack[64]. **b–d** Proxy-derived $pCO_2$ estimates (2.6–0.8 Ma) and ice core data (0.8–0 Ma)[36]. Also shown is the blue ice record at ca. 1 Ma[56]. Error bars are $1\sigma$ uncertainties. **b** Alkenone-derived $pCO_2$ records (red triangles (ODP 925)[17]; green and gray triangles (ODP 806 and 1208)[50]; yellow triangles (ODP 999)[51]). **c** Boron-derived $pCO_2$ records (yellow circles (ODP 668)[49]; green circles (ODP 999)[51]; gray circles (ODP 999)[52]; red circles (ODP 999)[48]). Recently published high-resolution $pCO_2$ records across the MPT[62] are shown as the dark blue curve, with a lighter gray band representing the full 95% envelope of the sampled distribution. **d** Luochuan paleosol-based $pCO_2$ estimates. Horizontal dashed lines indicate the pre-industrial $CO_2$ level (280 ppm)

necessarily always hold. In addition, the boron method requires the knowledge of a second carbonate system such as alkalinity or dissolved inorganic carbon[54]. Similarly, the alkenone method requires constraints on the physiological conditions of alkenone-producing haptophyte algae[55]. Curiously, the recently studied blue ice which are outcrops of old ice from Antarctica show that $pCO_2$ remained <300 ppm for at least 1 million years[56] and possibly 2.7 million years[57], contradicting to some of the boron-based and alkenone-based results (Fig. 5b, c). In this respect, terrestrial archives free of the air–sea disequilibrium of $CO_2$ problem could weigh into the absolute values of Pleistocene $pCO_2$ problem.

Our terrestrial-based record shows that interglacial $pCO_2$ levels during 2.6–0.8 Ma varied between 183 and 292 ppm (averaged for each interglacial), with the lower $pCO_2$ levels within this range often associated with samples collected close to the paleosol–loess (i.e. interglacial–glacial) transitions. These $pCO_2$ estimates are similar to the ice core $pCO_2$ record of the last 800 ky (Fig. 5d), and consistent with the blue ice records which show <300 ppm $pCO_2$[56].

Coupled general circulation model/ice sheet model have suggested that, the $pCO_2$ has to drop below the threshold of 280 ppm to induce major Northern Hemisphere glaciations (NHG)[58]. This is confirmed by this paleosol-based record, as well as a recently published high-resolution $\delta^{11}B$-$pCO_2$ record[48]. However, some of the earlier estimates using marine archives showed >400 ppm of $pCO_2$ estimates prior to ca. 2.3 Ma[48,50] (Fig. 5b, c). A recent study has identified that the traditional alkenone method tends to

overestimate Pleistocene $CO_2$, which can be explained by the physiological parameter $b$ used that are on average, too large[59].

The gradual decline of $pCO_2$ over the last 3.0 Ma has been hypothesized to trigger the MPT transition[60,61]. Recently published high-resolution $pCO_2$ record using $\delta^{11}B$ proxy[62] documents an overall higher $CO_2$ level ($241 \pm 21$ ppm) during the early part of the MPT (1.25–1.08 Ma) than that during 0.26–0 Ma ($203 \pm 14$ ppm) (Fig. 5c). Our paleosol-based $pCO_2$ values also demonstrate a statistically significant decline of ~40 ppm from the MPT to the post-MPT period (Fig. 4a). However, the $pCO_2$ record presented in this study shows that the MPT itself was accompanied by a $CO_2$ rise of ~50 ppm (Fig. 4a). Calculation of factor changes in $CO_2$ among three time periods (i.e. pre-MPT, MPT, and post-MPT) also show a significant $CO_2$ peak during the MPT, and no apparent $CO_2$ difference before and after the MPT (Fig. 4). This analysis indicates that the low $CO_2$ condition was already established long before the MPT. As a consequence, our results do not support the supposition that the decline of $pCO_2$ lead to the onset of the MPT.

## Methods
**Chronology**. The chronological framework of loess–paleosol sequences in Luochuan are based on the paleomagnetic reversal sequences, with the detailed information described in previous studies[63]. The Brunhes–Matuyama reversal occurs among the transition of L8 (loess unit 8) and S8 (paleosol unit 8). The Jaramillo subchron is defined between the upper part of L10 and the base of S11, while the Olduvai subchron lies between the middle part of L25 and the lower part of S26. A basal age of ~2.6 Ma is estimated for the Chinese loess deposits, based on

the Matuyama/Gauss magnetic reversal occurring in the oldest loess unit, L33. The stratigraphy of loess–paleosol sequences can be further delineated by MS data, which was measured through a Bartington MS system (5% accuracy), with higher bulk MS occurring in paleosols and lower values in loess layers. The good correlation between MS of loess–paleosol sequences and glacial–interglacial stages recorded by benthic $\delta^{18}O$ (Supplementary Fig. 1) further supports the robustness of this chronology. To be able to directly compare between paleosol-derived and marine-derived $p$CO$_2$ records, we tuned the MS data to the benthic $\delta^{18}O$ stack LR04[64]. Specifically, the top and basal age of each paleosol unit is assigned by the beginning and end of the corresponding interglacial episode (defined by the higher 50 percentiles of $\delta^{18}O$ values), and the ages within a single unit were calculated through interpolation. Because of the shallow depth of origin of the pedogenic carbonates, we did not apply any offset between their ages and the ages of paleosol layers where they were sampled from.

**$\delta^{13}C$ and trace element analyses**. Calcite and dolomite contents of bulk soil samples were measured through Fourier transform infrared spectrophotometry, following procedures described in ref. [24]. To ensure that our samples were free of detrital carbonates, only paleosols without dolomite were applied for subsequent analyses[24]. In order to obtain $\delta^{13}C_r$ and $\delta^{13}C_s$ for $p$CO$_2$ calculation, the $\delta^{13}C$ values of pedogenic carbonates in bulk soil samples (i.e. FDC), and co-occurring SOM were measured. For carbonate isotope analyses, bulk soil samples were treated with 10% H$_2$O$_2$, the remaining of which were then oven-dried, crushed, and grinded into powder for homogenization. Samples were then analyzed on a Thermo-Finnigan MAT 253 isotope ratio mass spectrometer using a Kiel IV carbonate device (75 °C reaction in 100% H$_3$PO$_4$). For organic carbon isotope analyses, bulk soil samples were reacted with 1 M HCl to ensure complete dissolution of carbonates, and then ultrasonically rinsed with deionized water for multiple times until pH reached neutral. The oven-dried residues were grinded into powder, loaded into tin capsules and analyzed on the ThermoFinnigan MAT 253 isotope ratio mass spectrometer with a Costech elemental analyzer attached. Both carbonate and organic carbon isotopic results are reported in permil (‰) notation relative to the Vienna Pee Dee Belemnite (VPDB) standard with a precision better than ±0.2‰ from duplicate analyses. Multiple samples (>3) from each paleosol unit were measured, with the averaged standard deviations for each paleosol unit about 0.47‰ (carbonate) and 0.43‰ (SOM). For trace element analysis of carbonate fractions, bulk samples were dissolved in 0.2 M acetic acid following the method described in ref. [65]. Trace element concentrations of dissolved solutions were then measured on an ICP-OES 6300. Precisions are better than 0.3% for Mn/Ca, Mg/Ca, and Sr/Ca following ratio calibration method described in ref. [33].

**Input parameters and error propagation**. In this work, input parameters related to the paleobarometer equation were either directly measured or estimated. $\delta^{13}C_s$ values can be solved by $\delta^{13}C_c$ through a temperature-dependent fractionation equation[8]. Previous work based on Pliocene pedogenic carbonates from the CLP[66] revealed that the formation temperature of pedogenic carbonates is $1-2$ °C lower than modern day summer air temperature (from June to September), and since Pliocene was globally warmer than Pleistocene[67,68], the formation temperature of pedogenic carbonates during the Pleistocene were probably not higher than the Pliocene. Therefore, we used a correction of $-1.5$ °C for the modern summer (JJAS) air temperature (18.4 °C for Luochuan), to represent the formation temperature of our pedogenic carbonates throughout the study interval. Uncertainty associated with temperature was set to ±3 °C. Previous investigations have determined minor $\delta^{13}C$ fractionation between soil-respired CO$_2$ and soil organic matter (SOM)[9], therefore $\delta^{13}C_r$ values were approximated through $\delta^{13}C$ of SOM from the same paleosol layer. However, carbon isotope fractionation occurs during decomposition of SOM, which could enrich the $\delta^{13}C$ of bulk paleosol organic matter[69]. Therefore, we applied a universal correction of $-1 \pm 0.5$‰ for our $\delta^{13}C_{SOM}$, following the difference of $\delta^{13}C$ values between the A and B horizon in an archived soil[70]. Errors associated with the $\delta^{13}C_s$ and $\delta^{13}C_r$ were assigned based on the standard deviations of measured $\delta^{13}C_c$ and $\delta^{13}C_{SOM}$ ($n > 3$) within the same paleosol unit. $\delta^{13}C_a$ values were derived from measured $\delta^{13}C$ values of contemporaneous marine benthic foraminifera carbonates[10]. To calculate $p$CO$_2$ and propagate errors associated with input variables, we used the MatLab-based PBUQ program[7]. PBUQ calculates probability density functions for each variable based on input parameters and associated uncertainties (assumed to be normal distribution), which are generated using 10,000 iterations (Monte Carlo simulations). This program yields median $R$ and $p$CO$_2$ values with error bars extending to the 16th and 84th percentiles.

## Data availability

All stable carbon isotope and magnetic susceptibility data from the Luochuan section of the Chinese Loess Plateau, as well as the calculated $S(z)$ and $p$CO$_2$ are attached as a Supplementary Data file.

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

## Acknowledgements

This work was funded by National Natural Science Foundation of China (41773118 and 41230526 to J.J.) and the program A for Outstanding Ph.D. candidate of Nanjing University (to J.D.).

## Author contributions

J.J. and J.D. conceived this research. J.D. and X.M. performed laboratory analyses. J.D. and G.L. performed statistical analysis of the data. J.D., Y.G.Z., and G.L. wrote the manuscript with input from all authors. All authors contributed to the interpretation of the data, led by J.J. and Y.G.Z.

## Competing interests

The authors have no competing interests to declare.
