## [Peer Review File · Nature Communications]

Reviewers' comments:

Reviewer #1 (Remarks to the Author):

I reviewed this manuscript previously for another journal where I thought it deserved to be published. Likewise, I think the manuscript deserves to be published in NCOMMS, for the following reasons. The conclusions are important and the approach is both novel and, for the most part (see comments on error propagation), well-supported by data. The work is at the forefront of paleoatmospheric CO₂ research, both from the perspective of proxy development and from the perspective of application to an important scientific question: "was the mid Pleistocene transition driven by a change in CO₂?" Crucial here is that the ONLY way we have to know whether our paleo CO₂ reconstructions are accurate is by comparing with other paleoCO₂ reconstructions. Therefore, we absolutely need multiple records from independent proxies that span the same time periods. When the result is disagreement, we typically learn something about the proxies. This work provides that comparison and at the same time addresses an important scientific question.

I have a few concerns, mostly with error propagation that I detailed below. It is very important that errors are reported as accurately as possible. I think some of the marine proxies underestimate their true error so I am not picking on this particular proxy here. But error bars are only as useful as they are honest. I would stress that EVEN if the error bars on absolute magnitude of CO₂ are increased once the authors inspect their error propagation calculations that the error on secular CHANGES in CO₂ are probably no so large as to invalidate the conclusions drawn here. See Ji et al 2018 EPSL A symmetrical CO₂ peak and asymmetrical climate change during the middle Miocene for an example of the strength of considering relative changes in CO₂ determined using paleosol carbonates.

I did not look carefully at the climate sensitivity analysis as it seems to me that not much can be learned about climate sensitivity from these new data, which is confirmed by the authors' conclusion that the calculated range of climate sensitivity is within the known range. So I don't think that part of the manuscript adds much. Nonetheless, I think the rest of the manuscript stands on its own and should be published in NCOMMS after revision.

Line by line and specific comments:

Line 32. Please do not confuse concentration and partial pressure. The units are different. More importantly, one varies with altitude (partial pressure) and the other does not (concentration). This is critical to the full understanding of the effect of changing CO₂ on climate.

Line 138

Please also provide the 2.5 and 97.5 percentiles

Fig1b the CO₂ from paleosol seems to be biased low, which I don't understand since this is the calibration dataset. Please explain.

lines 181-190. For a balanced discussion, please explain some of the issues with using finely disseminated carbonates and how you mitigate them. How do we know these carbonates weren't reset (or even reformed) while in the modern exposure surface. How do we know they don't all record modern atmospheric CO₂? Where in the soil (e/g/what depths) does it form? Which MS measurements are used to compared with the measurements of the disseminated calcite? Is there no disseminated calcite in the glacial loess? Why didn't you reconstruct glacial CO₂?

How did you sample the disseminated carbonates/ separate them from bulk loess and/or paleosol

material for analysis?

Line 198, specify that you mean atmospheric pCO₂ here.

199 what does 'shallow vadose zone' mean. This could mean top 5 cm or top 2 m.

Line 219- then why make all this fuss about NOT using nodules? Do you get much different CO₂ concentrations if you do everything the same but instead use the nodule data from figure S4? Are the 'R' values much lower if you use nodules- I think they probably are if so, you should say this. Another thought- if the nodules formed deeper, then can you use nodules and disseminated calcite together to approximate the soil CO₂ profile? This might add constraints to CO₂ determinations.

Lines 263-264. Secular changes in B isotope composition of seawater should be slow. Please support this claim with e.g., evidence for turnover times

Line 300. Where does this error bar on your estimate of MPT CO₂ come from? From the data on figure 3 the error seems to be much larger. Do these error bars consider all known sources of uncertainty, including uncertainty associated with the regression of S(z) on MS (Figure 1a). If it does include this uncertainty, please remember that the error should be calculated for a new estimate of S(z) (i.e. uncertainty with which a new measurements of MS can be used to determine a value of S(z)). THIS IS NOT THE SAME AS using the standard error of the regression line (which represents the error on the slope and intercept of the regression line itself). There is a difference between the uncertainty associated with the regression line itself (i.e. 'sampling variance' in other words if you took another set of random samples, how different would the regression line be) and the uncertainty associated with using a regression line and a new observation (in this case MS) to determine the value of the unknown (in this case S(z)). The latter is sometimes referred to as sampling variance for a new observation and is what you need to calculate in order to propagate error associated with the regression. For typical 'Y from X' regressions in which the independent variable is measured to determine the value of the dependent (or the response) variable, the assumption in the regression analysis is that all the error is associated with Y (i.e. the value of X is known perfectly). This case does not entirely apply to calculating S(z) from MS, because MS does not control S(z) (nor does S(z) control MS, rather both variables are likely controlled by , e.g. precipitation). However when creating the S(z) versus MS regression curve, most of the error is probably in S(z) - the MS is measured directly and very well-known. Therefore, the 'Y from X' approach is probably a good approximation of error. When I did this for PBUQ, I followed Davis 2002 (as cited in Breecker 2013 G-cubed). The relevant equations are in the PBUQ code. If you propagate error, it should be done correctly (which, unfortunately, is actually pretty rare). I say all this because the reported errors look too small to me. But I did not go through the propagation myself.

Lone 308. This may not be true. The MPT could be related to changes in glacial CO₂ (which the present manuscript does not address).

It is hard for me to believe that the interglacial CO₂ reported here provide any useful constraint on climate sensitivity. The error on CO₂ is rather large and the temperature difference among interglacials is rather small.

Please provide the MS values in the supplementary tables (cited appropriate references if not newly measured here) so that readers know exactly which values were used in conjunction with the new d¹³C values presented here.

What temperature did you use to calculate the d¹³C value of soil CO₂ from measured d¹³C values of

soil carbonate. Please put these temperatures in the supplementary table along with an estimate of the uncertainty on the formation temperature of the disseminated carbonates. Please propagate the uncertainty on temperature through to the calculated CO₂.

review by Dan Breecker

Reviewer #2 (Remarks to the Author):

In this manuscript Da et al propose that the paleosol paleobarometer for atmospheric CO₂ reveals low values through the Pleistocene based on paleosols in the Chinese Loess Plateau. The result is uncontroversial and uncontested as the paper is constructed, but there is a more important novel contribution buried in the details. Thus I recommend rejection as it stands, but reformulation.

The really novel aspect of this work is the contention that paleosol productivity can be approximated by magnetic susceptibility. This is not apparent in the abstract but only in the text, which is a major shortcoming. More serious though is that use of magnetic susceptibility as a paleoproductivity proxy is not firmly established by observation that magnetic susceptibility increased from west to east on the loess plateau, because magnetic susceptibility is also a function of time for soil formation. Time for formation could have been constrained by nodule size for example, but no such constraint has been done. Furthermore there are no observations of actual soil productivity, unlike the cited studies using mean annual precipitation and depth to carbonate which are based on field studies of carbon dioxide in soil. How exactly are the productivity S_{9Z0} values "back calculated" (l.129)? Because time for formation was not considered, such back calculation must be flawed and is perhaps circular reasoning. Presumably some modern observations of soil productivity are included somehow, but there is not mention of them. I think the use of magnetic susceptibility for paleoproductivity is very promising and should be a first paper in this series, not buried on the way to other problems.

The critical "back calculation" is only explained in the supplementary material and is actually based on the assumption that the ice core data is correct. Thus this new paper is just a reiteration of that data, and not an independent assessment. At the very least this should be spelled out in the paper.

What is clearly needed is a database of magnetic susceptibility and carbon dioxide contents of modern soils on loess in China.

Contrary to line 88, the reference cited does not question the applicability of MAP or Bk depth to these equations

CLP in l. 102 Please spell out Chinese Loess Plateau

l.202 Needle fiber calcite is considered a fungal precipitate, and may not be representative of bulk carbonate in isotopic composition. Wright, V.P. (1986) The role of fungal biomineralization in the formation of early Carboniferous soil fabrics. *Sedimentology* 33, 831-838.

Mismatches between boron or alkenone and the paleosol paleobarometer are not really a surprise considering errors. Also the sensitivity relationships are weak.

Reviewer #3 (Remarks to the Author):

Da and colleagues present a record of atmospheric CO₂ for parts of the Pleistocene. Most existing records of CO₂ with a high temporal resolution come from marine-based proxies (e.g., boron and alkenone methods). This study is novel in that it uses the (terrestrial) paleosol carbonate proxy.

I'll start my review with some larger concerns:

1) Even if there is no detrital carbonate present in these soils today (and the authors do not make a convincing case), their presence back when the soils were actively developing is concerning because their dissolution and/or re-precipitation would affect S(z) and the δ¹³C of CO₂ in the pore space. In

other words, the authors are dealing with a three-end-member mixing model. The original developers of the paleosol-CO₂ method were clear that soils with detrital carbonate should be avoided.

2) This is the first study I am aware of that targets carbonate from shallow soil depths. The authors point out why this is dangerous—the $S(z)$ value has a strong vertical gradient at shallow depths (lines 89-92). That is, small differences in depth can correspond with very different $S(z)$ values. This has been documented repeatedly in modern soils. The authors fail to address this shortcoming. Why should we trust their estimates of $S(z)$ if it is highly sensitive to small differences in soil depth?

3) On the topic of $S(z)$, the inverse calculation of $S(z)$ for the 800-0 kyrs interval assumes no biases in the other inputs. In other words, the reported percentiles based on the resampling routine (columns L and M in Table S2) are a gauge of precision, but not accuracy.

4) It is misleading to compare CO₂ estimates to the ice-core record (lines 137-143 and Figure 1b) because the estimates of $S(z)$ used to calculate CO₂ are based on...the ice-core CO₂ record. This is circular logic. A more compelling approach would be to create a MS- $S(z)$ regression from a subset of data, and then apply the regression to estimate CO₂ with the other, unused data. This could be done repeatedly, with different data subsets.

5) Is there an independent record of rainfall for your sequence (line 134)? It's surprising to me that the interglacials younger than ~1 Ma are wetter than the interglacials older than 1 Ma (as implied by the MS record presented in Figure S1).

6) The provocative part of this paper is in the title: low CO₂ throughout the Pleistocene. I am not convinced that the authors' record is different than the boron-based CO₂ records. The high CO₂ estimates from the boron method (Figure 3b) do not overlap in time with any samples from the current study (Figure 3c), with the possible exception of the three data points at 2.57 Ma. If you start at 2.32 Ma, the boron and paleosol records of CO₂ are essentially the same. The alkenone record is different, yes, but there are methodological reasons for this that the authors point out in the manuscript.

7) There is some evidence for higher Earth-system sensitivity during the Plio-Pleistocene (e.g., Royer 2016 Annual Review of Earth and Planetary Sciences). Also, the similar slopes between Figures 4b and 4c does not make sense. There were some continental ice-sheets at this time, so the slope of the red (and blue) line in Figure 4b should be steeper than that in Figure 4c.

Other comments:

Title is misleading—you are only looking at interglacials

Line 41: It's not clear which version of climate sensitivity you are talking about there (with or without land-ice feedback)

Line 57: Citations needed.

Line 59: "Eras"

There is a disconnect between these two statements: lines 122-124: "Median $S(z)$ levels over the last 800 ky range from 396 ppm to 943 ppm, with a standard error of +125/-93 ppm on average. The $S(z)$ estimates are consistent with $S(z)$ ranges defined by Holocene mollisols and aridisols (10) - the soil orders that our samples belong to." Lines 200-201: "back-calculated $S(z)$ values over the last 800 ky using our paleosol samples are significantly lower than previous results (10, 13) and closer to pCO₂ levels" I would expect your $S(z)$ values to be lower simply because your samples come from shallow

soil depths; so, I find the statement on lines 122-124 confusing.

Lines 187-193: the problems of translocation and detrital carbonate also apply to bulk carbonate from shallow soil depths (the statement on lines 228-229 about translocation is not backed up with evidence). It is unbalanced to call these issues a problem for carbonate nodules but not for disseminated bulk carbonate.

Line 242: The Pleistocene is not an "era".

Line 264: The "sudden" decline in the boron-based CO₂ estimate is probably an across-study artifact (the high estimates all come from one study—the red dots in Figure 3b).

Figure 3: Why are the estimates from Martinez-Boti (ref. 55) included in Figure 4 but not in Figure 3? Similarly, why are estimates from refs. 45, 50 and 52 included in Figure 3 but not in Figure 4?

Tables S1 & S2: Magnetic susceptibility needs to be included.

Supplement, line 97: why was +/- 3 oC chosen?

Author replies to reviewer comments on Da et al. ‘*Low CO₂ levels of the entire Pleistocene Epoch*’ [Paper# NCOMMS-19-00418]

The reviewers’ comments are in *blue and italic*; our replies are in black.

Reviewer #1 (Remarks to the Author):

Review by Dan Breecker

I reviewed this manuscript previously for another journal where I thought it deserved to be published. Likewise, I think the manuscript deserves to be published in NCOMMS, for the following reasons. The conclusions are important and the approach is both novel and, for the most part (see comments on error propagation), well-supported by data. The work is at the forefront of paleoatmospheric CO₂ research, both from the perspective of proxy development and from the perspective of application to an important scientific question: “was the mid Pleistocene transition driven by a change in CO₂?” Crucial here is that the ONLY way we have to know whether our paleo CO₂ reconstructions are accurate is by comparing with other paleoCO₂ reconstructions. Therefore, we absolutely need multiple record from independent proxies that span the same time periods. When the result is disagreement, we typically learn something about the proxies. This work provides that comparison and at the same time addresses an important scientific question.

We thank Dr. Dan Breecker for spending time and effort reviewing our manuscript for multiple times. Dr. Breecker views on our work positively as ‘deserves to be published in NCOMMS’, ‘novel and for the most part well-supported by data’, and ‘at the forefront of paleoatmospheric CO₂ research both from the perspective of proxy development and from the perspective of application to an important scientific question’. We sincerely appreciate the encouragement and acknowledgement from Dr. Breecker, a leading expert in the field of paleoatmospheric CO₂ research. We totally agree with the reviewer that the quality of our paleo-CO₂ reconstructions must be examined via comparing to other independent proxy-based records, and that the agreement/disagreement between different proxy systems provides key information about the proxies themselves. This is exactly the reason that we compared CO₂ among multiple records, and we really appreciate that that the reviewer saw the value.

Dr. Breecker had several constructive and thoughtful comments, which were mainly focused on (1) proper ways of error propagation, (2) sample quality (e.g. diagenetic influences), and (3) the necessity of the climate sensitivity analysis. Those comments have been really helpful guiding us to revise the manuscript. To address these comments and concerns, we have done a substantial amount of revisions, including experimental, statistical and modeling work, as detailed below in the point-to-point responses and in the main text. We summarize our revisions in response to the three major comments here:

(1) for error propagation, we want to make it clear that in the previous submission we have carefully calculated the errors on $S(z)$ and reconstructed atmospheric $p\text{CO}_2$, taking into account both the errors on the parameters themselves and the errors on the MS- $S(z)$ relationship (i.e. the errors on the slope and intercept of the least-square fitting line); we did the calculation by using the PBUQ program and Monte Carlo random sampling simulations to generate a population of $p\text{CO}_2$ results; to clarify, Dr. Breecker has made this comment in the previous round of review in another journal, and we have followed his suggestions and re-done the error analysis in this version; we now expanded the relevant text to better illustrate how we estimate errors;

(2) to address the comment on sample quality, we have done substantial new analyses, including measurement of trace element and SEM analysis of the carbonate fractions in the paleosol samples, to demonstrate the pedogenic origin and minimal diagenetic influence for our studied samples;

(3) we have removed the climate sensitivity analysis as suggested by the reviewer, to make the main text succinct and more focused on our new method and relevant findings.

Overall, these revisions do not fundamentally change our conclusions, and we do feel the manuscript is greatly strengthened after incorporating these revisions. We hope the reviewer now agrees we address his concerns. Please see our detailed replies below.

I have a few concerns, mostly with error propagation that I detailed below. It is very important that errors are reported as accurately as possible. I think some of the marine proxies underestimate their true error so I am not picking on this particular proxy here. But error bars are only as useful as they are

honest. I would stress that EVEN if the error bars on absolute magnitude of CO₂ are increased once the authors inspect their error propagation calculations that the error on secular CHANGES in CO₂ are probably no so large as to invalidate the conclusions drawn here. See Ji et al 2018 EPSL A symmetrical CO₂ peak and asymmetrical climate change during the middle Miocene for an example of the strength of considering relative changes in CO₂ determined using paleosol carbonates.

We thank the reviewer for emphasizing the importance of doing error propagation in a correct way. We totally agree that error propagation must be done correctly to make error bars 'honest' and 'useful'. As mentioned, we have followed Dr. Breecker's suggestion (taking into account both the errors on the parameters and the errors on the MS-S(z) regression) and rigorously propagated errors on the estimated $p\text{CO}_2$. We now expanded the relevant text on technical details and emphasized that we did account for both the errors on the parameters and on the MS-S(z) regression.

Motivated by the reviewer's comment on the changes in CO₂ and errors, we also look at the relative change of CO₂ rather than only looking at the absolute magnitude of CO₂. We adopted the method of calculating "factor change in CO₂" to better illustrate the variations of $p\text{CO}_2$ across the Mid-Pleistocene transition (MPT) period. In this factor, we take into account the variability in reconstructed $p\text{CO}_2$ (taking the mean values and standard errors of multiple CO₂ data for a given time period) and report the resulted probability distributions of relative $p\text{CO}_2$ changes over time, providing a statistically more robust approach to evaluate CO₂ variations. We have added two paragraphs (Line 286-300, 375-387) and a new figure (Fig. 4) in the new manuscript to discuss CO₂ changes across the MPT.

I did not look carefully at the climate sensitivity analysis as it seems to me that not much can be learned about climate sensitivity from these new data, which is confirmed by the authors' conclusion that the calculated range of climate sensitivity is within the known range. So I don't think that part of the manuscript adds much. Nonetheless, I think the rest of the manuscript stands on its own and should be published in NCOMMS after revision.

We thank the reviewer for this suggestion. This comment echoes the points made by the other two reviewers. To make our manuscript succinct and to avoid potential confusions for readers, we have now removed the part related to the discussion of climate sensitivity and focused on the rest part on the novelty of our approach and the importance of our findings.

Comment 1. Line 32: Please do not confuse concentration and partial pressure. The units are different. More importantly, one varies with altitude (partial pressure) and the other does not (concentration). This is critical to the full understanding of the effect of changing CO₂ on climate.

We thank the reviewer for pointing it out. We now use the abbreviation “*p*CO₂” to represent “the partial pressure of atmospheric CO₂” throughout the new manuscript, and have clarified this in the main text (Line 32).

Comment 2. Line 138: Please also provide the 2.5 and 97.5 percentiles.

Done. We have added the 2.5 and 97.5 percentiles of *p*CO₂ distributions in the supplementary table.

Comment 3. Fig1b: The CO₂ from paleosol seems to be biased low, which I don't understand since this is the calibration dataset. Please explain.

We thank the reviewer for this comment. It is likely that our CO₂ estimates in the original figure are visually biased. Statistically, the mean CO₂ level of our CO₂ estimates during the last 800-ky interglacial cycles is 252 ppm ($1\sigma = +103/-91$ ppm), similar to that derived from the ice core data (246 ± 15 ppm). Nonetheless, to avoid circular reasoning, we now adopted a resampling method when validating the reconstructed CO₂ in comparison to ice core data, as suggested by Reviewer #3. Specifically, we divided our 800-ky paleosol samples ($n=22$) into two subsets—a training sample group and a test sample group. We then establish a MS-S(*z*) regression from the training subset, use it to calculate the S(*z*) and related *p*CO₂ for the test sample set, and compare the *p*CO₂ to the ice core data. We vary the number of samples in the training group from $n = 10$ to $n = 21$. For a given training sample number *n*, we perform a bootstrap sampling for 1,000,000 times. During each run, we calculate the mean relative difference χ between the calculated *p*CO₂ and those from the ice core data. The χ distributions resulted from 1,000,000 iterations centered around 0, with >70% data points falling within 10% difference. We show this in Lines 242-257, Fig.1b and Fig. S5 of the new manuscript.

Comment 4. lines 181-190. For a balanced discussion, please explain some of the issues with using finely disseminated carbonates and how you mitigate them.

How do we know these carbonates weren't reset (or even neoformed) while in the modern exposure

surface?

How do we know they don't all record modern atmospheric CO₂?

Where in the soil (e/g/what depths) does it form?

Which MS measurements are used to be compared with the measurements of the disseminated calcite?

We thank the reviewer for this insightful comment concerning the issues with finely disseminated carbonates. These comments are mainly focused on the quality and potential alterations of our studied carbonate fractions and their validity in reconstructing past $p\text{CO}_2$. We have done substantial revisions to address these concerns. Guided by the questions asked, we have expanded our discussion to argue the purity and quality of finely disseminated carbonates in bulk paleosols as a paleoclimate archive (Lines 132-186). Besides the questions the reviewer asked (i.e. diagenetic influence, possible noise from modern atmospheric CO₂, sampling depth, details about MS measurement), we think another important point we want to emphasize, to complement the reviewer's comments, is that our studied carbonate fractions (finely disseminated carbonates, FDC) were formed during pedogenesis, and do not contain detrital carbonate inherited from parent material. In our response below, we first explain how we reject the possible contamination from detrital carbonates, and then then provide a detailed point-to-point reply in response to each specific question.

As said, we render that a major problem related to the finely disseminated carbonates (FDC) in bulk paleosol samples is the potential contamination of detrital carbonates inherited from initial parent material. To circumvent this issue, we only select bulk paleosols with no occurrence of dolomite minerals. This is because soil water preferentially dissolves calcite over dolomite¹, and since dolomite cannot form during pedogenesis in the loess-paleosol sequence thus are purely inherited, the disappearance of dolomite suggests complete dissolution of detrital carbonate². As mentioned in the previous manuscript, several lines of evidence have suggested minimal contribution of detrital carbonate:

- 1) scanning electronic microscopic analysis shows that FDC are mainly composed of needle fiber shaped calcites, which is of a pedogenic origin³;
- 2) the long-term identical trends of $\delta^{13}\text{C}$ values between FDC and coeval calcite nodules — typical pedogenic carbonates.

To provide more evidence for the pedogenic origin of our paleosol samples, we performed analysis on the trace element concentrations of the carbonate fractions in bulk paleosol samples. The trace elemental

ratios (Sr/Ca, Mg/Ca and Mn/Ca) are significantly different from those of detrital carbonate from potential source regions⁴, and similar to those of the microcodium—another kind of authigenic carbonate in the CLP⁵, which further confirmed the neglectable contribution of detrital carbonate in our bulk paleosols. We have added a new Figure (Fig. 1e and 1f) to present the results of trace elemental ratios, and discussion of the results in Lines 159-170.

Given the multiple concerns raised, we next replied to each specific question.

‘How do we know these carbonates weren’t reset (or even neoformed) while in the modern exposure surface?’, and *‘How do we know they don’t all record modern atmospheric CO₂?’*

We are aware that it is crucial to avoid pedogenic carbonates formed under modern conditions. Before field sampling, we trenched the soil profiles (>1 m deep) to ensure a fresh exposure, and monitored the magnetic susceptibility (MS) values of bulk paleosols, which were compared to published data. Moreover, the general trends of $\delta^{13}\text{C}$ between FDC in this study and coeval calcite nodules are indistinguishable from each other (Fig. S4), and our back-calculated S(z) using FDC over the last 800 ky are well correlated with MS values of bulk paleosol samples (Fig. 1a), which further eliminates the possibility of inappropriate sampling. To better illustrate this, we have added several more sentences in Lines 126-128 to explain the sampling routines.

‘Where in the soil (e/g/what depths) does it form?’

Due to the aggradational nature of loess, the exact formation depth of FDC is difficult to pin down directly. Nonetheless, multiple pieces of evidences indicate a shallow formation of FDC: i) the FDC were identified in the soil Bt/Bw horizons upon field observations; ii) needle fiber calcite—the major carbonate micromorphology of FDC, mostly appears in the transition zone between humic horizon and subsurface C zone³. Detailed discussion of its formation depth has been added in Lines 139-142.

‘Which MS measurements are used to be compared with the measurements of the disseminated calcite?’

MS and $\delta^{13}\text{C}$ values measured from the same bulk paleosol samples were used for comparison.

Comment 5. Is there no disseminated calcite in the glacial loess? Why didn’t you reconstruct glacial CO₂?

Pedogenic carbonates were also formed in the loess units during glacials. However, all the glacial loess samples contain significant amount of detrital carbonate, suggested by the presence of dolomite, which make them inappropriate for $p\text{CO}_2$ reconstruction.

Comment 6. How did you sample the disseminated carbonates/ separate them from bulk loess and/or paleosol material for analysis?

First of all, to avoid contamination of regolith, we trenched the soil profile, and monitored the MS values which were compared to published data before sampling. For subsequent analyses, we only choose bulk paleosol samples from the paleosol units (i.e. interglacial episodes) with no dolomite minerals. Based on the sizes of soil carbonates (i.e. nm- μm) revealed by SEM photography, we term these carbonate fractions as finely disseminated carbonates. Detailed sampling routines have been provided in Lines 126-128.

Comment 7. Line 198, specify that you mean atmospheric $p\text{CO}_2$ here.

Done.

Comment 8. Line 199 what does 'shallow vadose zone' mean. This could mean top 5 cm or top 2 m.

We thank the reviewer for this comment. With more careful consideration, we acknowledge that this term 'shallow vadose zone' may be obscuring whereas the formation depth of FDC is not directly measured in the field, but is constrained from field observations and other mineralogical evidences. To clarify, we now remove the term "shallow vadose zone", and provide a more detailed discussion to explain the location of FDC at Lines 139-142.

Comment 9. Line 219 then why make all this fuss about NOT using nodules? Do you get much different CO_2 concentrations if you do everything the same but instead use the nodule data from figure S4? Are the 'R' values much lower if you use nodules- I think they probably are if so, you should say this.

Another thought- if the nodules formed deeper, then can you use nodules and disseminated calcite together to approximate the soil CO_2 profile? This might add constraints to CO_2 determinations.

Good suggestions, thanks. Unfortunately, we didn't perform analysis on calcite nodules in Luochuan section, and the nodule data presented in Fig. S4 are from other sections⁶, which are only used here to prove that FDCs used in this paper did not experience diagenesis during the depositional processes. We

now abandoned the part of comparison between nodules and disseminated calcite, and focus more on the characteristics of disseminated calcite in the new manuscript (Lines 132-186).

We also thank the reviewer for providing a great idea (i.e. using disseminated calcite and nodule data to reconstruct soil CO₂ profile), which is definitely a promising future research direction. We see this as a follow-up work, whereas in this study we want to focus on disseminated calcite.

Comment 10. Lines 263-264. Secular changes in B isotope composition of seawater should be slow. Please support this claim with e.g., evidence for turnover times

Agreed. We have changed the sentence to “the boron method requires the knowledge of a second carbonate system such as alkalinity or DIC” in Lines 343-344.

Comment 11. Line 300. Where does this error bar on your estimate of MPT CO₂ come from? From the data on figure 3 the error seems to be much larger.

We thank the reviewer for this careful catch. In the previous manuscript, the error bar (± 25 ppm) represents the standard deviation of all the $p\text{CO}_2$ estimates during MPT, which is significant lower than the standard errors (1σ) of individual $p\text{CO}_2$ estimates. To avoid confusion, in the new manuscript, we instead use the mean 1σ errors of all the $p\text{CO}_2$ estimates during the MPT, which is +105/-94 ppm (Line 297).

Comment 12. Do these error bars consider all known sources of uncertainty, including uncertainty associated with the regression of $S(z)$ on MS (Figure 1a). If it does include this uncertainty, please remember that the error should be calculated for a new estimate of $S(z)$ (i.e. uncertainty with which a new measurements of MS can be used to determine a value of $S(z)$). THIS IS NOT THE SAME AS using the standard error of the regression line (which represents the error on the slope and intercept of the regression line itself). There is a difference between the uncertainty associated with the regression line itself (i.e. ‘sampling variance’ in other words if you took another set of random samples, how different would the regression line be) and the uncertainty associated with using a regression line and a new observation (in this case MS) to determine the value of the unknown (in this case $S(z)$). The latter is sometimes referred to as sampling variance for a new observation and is what you need to calculate in order to propagate error associated with the regression. For typical ‘Y form X’ regressions in which the

independent variable is measured to determine the value of the dependent (or the response) variable, the assumption in the regression analysis is that all the error is associated with Y (i.e. the value of X is known perfectly). This case does not entirely apply to calculating S(z) from MS, because MS does not control S(z) (nor does S(z) control MS, rather both variables are likely controlled by, e.g. precipitation). However, when creating the S(z) versus MS regression curve, most of the error is probably in S(z) - the MS is measured directly and very well-known. Therefore, the 'Y from X' approach is probably a good approximation of error. When I did this for PBUQ, I followed Davis 2002 (as cited in Breecker 2013 G-cubed). The relevant equations are in the PBUQ code. If you propagate error, it should be done correctly (which, unfortunately, is actually pretty rare). I say all this because the reported errors look too small to me. But I did not go through the propagation myself.

We thank the reviewer for the detailed explanation and instruction on error propagation. We totally agree that it is critical to do error estimates in a correct way. In summary, it is true that our reported errors have included the errors associated with all the input parameters for the paleosol barometer equation by Cerling (1992)⁷. For the error on S(z), we applied Gaussian error propagation, which includes the standard errors of the slope and intercept of the MS-S(z) regression equation, as well as the errors on the new observation value of MS. We have added a subsection in the Methods section (Lines 431-452) to explain our procedures for propagating errors.

Comment 13. Line 308. This may not be true. The MPT could be related to changes in glacial CO₂ (which the present manuscript does not address).

Yes, but it's not supported by our currently available data, although we acknowledge the limitations of our data. Please see Lines 381-387.

Comment 14. It is hard for me to believe that the interglacial CO₂ reported here provide any useful constraint on climate sensitivity. The error on CO₂ is rather large and the temperature difference among interglacials is rather small.

This problem has also been raised by other reviewers, therefore we have removed the discussion for climate sensitivity in the new manuscript after consideration.

Comment 15. Please provide the MS values in the supplementary tables (cited appropriate references if

not newly measured here) so that readers know exactly which values were used in conjunction with the new $\delta^{13}\text{C}$ values presented here.

Done.

Comment 16. What temperature did you use to calculate the $\delta^{13}\text{C}$ value of soil CO_2 from measured $\delta^{13}\text{C}$ values of soil carbonate? Please put these temperatures in the supplementary table along with an estimate of the uncertainty on the formation temperature of the disseminated carbonates. Please propagate the uncertainty on temperature through to the calculated CO_2 .

Previous research using the clumped isotope thermometer have determined that the formation temperature of pedogenic carbonates across the CLP are generally 1-2 °C lower than modern summer air temperature (JJAS). Therefore, we applied a correction of -1.5 °C for the modern summer air temperature (JJAS) of Luochuan (18.4 °C) to represent the formation temperature of pedogenic carbonates used in this study. Moreover, sensitivity analysis (see Supplementary Information) shows that temperature exerts minimal impact on the calculated $p\text{CO}_2$. The temperature and its error has been added in the supplementary table and errors related to temperatures have been propagated.

Reviewer #2 (Remarks to the Author):

Comment 1. In this manuscript Da et al propose that the paleosol paleobarometer for atmospheric CO_2 reveals low values through the Pleistocene based on paleosols in the Chinese Loess Plateau. The result is uncontroversial and uncontested as the paper is constructed, but there is a more important novel contribution buried in the details. Thus I recommend rejection as it stands, but reformulation.

We thank Reviewer #2 for the constructive and insightful comments. We believe the conclusions drawn from this study are important to both the improvement of paleosol- CO_2 proxy, as well as the CO_2 evolution history during early Pleistocene. As mentioned in Reviewer #1's comment: "The only way we have to know whether our paleo- CO_2 reconstructions are accurate is by comparing with other paleo- CO_2 reconstructions. Therefore, we absolutely need multiple records from independent proxies that span the same time periods." We hope our replies convince the reviewer that this work is significant and deserves publication both the approach and the findings.

Comment 2. The really novel aspect of this work is the contention that paleosol productivity can be approximated by magnetic susceptibility. This is not apparent in the abstract but only in the text, which is a major shortcoming.

We appreciate the reviewer for acknowledging the novelty of our work. The application of magnetic susceptibility (MS) for paleosol productivity (S(z)) quantification has not been proposed before and is indeed a novel aspect of this work. To highlight this achievement, we have added “Using finely disseminated calcites precipitated in paleosols from the Chinese Loess Plateau, here we identify that S(z) associated with these calcites are low, and can be quantitatively constrained by soil magnetic susceptibility (MS).” in the abstract, and several paragraphs in the main text (Lines 214-233) to discuss the controlling mechanism of MS and its relation with S(z).

Comment 3. More serious though is that use of magnetic susceptibility as a paleoproductivity proxy is not firmly established by observation that magnetic susceptibility increased from west to east on the loess plateau, because magnetic susceptibility is also a function of time for soil formation. Time for formation could have been constrained by nodule size for example, but no such constraint has been done.

We acknowledge that before using any soil property for paleoclimate reconstruction, it is crucial to establish its behavior with time. Some properties develop in a linear fashion with time, and hence dependent on weathering duration, others evolved rapidly toward a near-steady-state equilibrium which is subsequently buried to form a paleosol, and hence preserved as a paleoclimate archive. We believe that it is the latter case with the soil magnetic susceptibility (MS) of the Chinese loess-paleosol sequence, based on the following evidences:

- 1) The ultrafine magnetite which dominates the MS signal of the loess-paleosol sequence⁸, is efficiently produced in-situ under well-drained soils with alternative wet and dry cycles;
- 2) Maher (1994)⁹ measured the MS values of young Holocene soils across the Chinese Loess Plateau (CLP), and found that they are in the same range as those of the buried paleosols, some of which have undergone much longer periods of pedogenesis;
- 3) Song et al. (2014)¹⁰ measured the MS values of 180 naturally vegetated surface soil samples across the CLP, the results of which display a clear southeast-northwest gradient, which is well correlated with modern mean annual precipitation (MAP).

In this case, both the observations and the scientific rationale behind supports MS as a rapidly forming

soil property which reaches steady-state equilibrium and documents ambient climatic conditions. Moreover, the steady accumulation rate of eolian loess during Quaternary interglacials¹¹ precludes the potential modification on MS through changes of dust accumulation rate. Therefore, MS has been widely used as a paleomonsoon indicator and here we believe its validity as a paleoproductivity proxy. We have added a paragraph (Lines 314-324) to address this issue in the new manuscript.

Comment 4. Furthermore, there are no observations of actual soil productivity, unlike the cited studies using mean annual precipitation and depth to carbonate which are based on field studies of carbon dioxide in soil. How exactly are the productivity $S(z)$ values “back calculated” (l.129)? Because time for formation was not considered, such back calculation must be flawed and is perhaps circular reasoning. Presumably some modern observations of soil productivity are included somehow, but there is not mention of them. I think the use of magnetic susceptibility for paleoproductivity is very promising and should be a first paper in this series, not buried on the way to other problems. The critical “back calculation” is only explained in the supplementary material and is actually based on the assumption that the ice core data is correct. Thus this new paper is just a reiteration of that data, and not an independent assessment. At the very least this should be spelled out in the paper. What is clearly needed is a database of magnetic susceptibility and carbon dioxide contents of modern soils on loess in China.

We agree that modern observations might provide more direct evidence in terms of the relationship between $S(z)$ and MS. This would be an interesting topic for a future study. Although measurements of $S(z)$ in modern soils are extremely scarce, we managed to gather a compilation of soil respiration rate (i.e. soil CO₂ flux at soil-air interface) based on published records¹²⁻¹⁸. These measurements were performed on loess-parented, naturally vegetated modern soils. As expected, the MS and soil respiration rate also demonstrate significant correlations in modern soils (Fig. 1), which further confirmed the relationship between MS and soil productivity, thus our MS- $S(z)$ approach based on paleosols in this study.

Fig. 1. Plots of MS versus soil respiration rate (SR) based on modern soil measurements. SR data were split into three groups—the mean annual group, the growing season (from June to September) group and the non-growing season group (from October to May).

For the current study, we added a new subsection (see Lines 188-212) to detail how the back-calculated $S(z)$ were determined. Moreover, the soil MS are statistically significantly correlated with back-calculated $S(z)$ (Fig. 2a in the main text). We also added several paragraphs to explain this correlation from a mechanistic point of view (see Lines 214-233). We conclude that this correlation is robust.

Comment 5. Contrary to line 88, the reference cited does not question the applicability of MAP or Bk depth to these equations.

We believe the reference¹⁹ raised certain doubt on their applicability on paleosols. As mentioned in Breecker and Retallack (2014): “The measurements of soil $S(z)$ in these studies were made during relatively short-duration (i.e. 1 or 2 growing seasons) and in some cases low temporal resolution (as low as only 1 growing season measurement) soil gas monitoring studies. If calcite primarily accumulates in soils during sporadic droughts (i.e. not every year)²⁰, then the gas monitoring studies used for calibration may not have captured carbonate accumulation events and the $S(z)$ values used for calibration may be overestimates. Alternatively, carbonate in some of the soils used for the paleoprecipitation and calcic-depth calibrations may have formed at a different time of year than was assumed. Both of these could result in inaccurately assigned $S(z)$ values, given that soil CO_2 concentrations vary seasonally and interannually (see compilation of seasonal variations in Ref^{20,21}).”

Comment 6. CLP in line 102 Please spell out Chinese Loess Plateau

Done.

Comment 7. Line 202: Needle fiber calcite is considered a fungal precipitate, and may not be representative of bulk carbonate in isotopic composition. Wright, V.P. (1986) The role of fungal biomineralization in the formation of early Carboniferous soil fabrics. Sedimentology 33, 831-838.

We thank the reviewer for this note. The origin of needle fiber calcite (NFC) is connected to both microorganic or inorganic processes as they may arise from either fungal biomineralization or from physicochemical precipitation from soil solutions²². Nonetheless, previous work has determined $\delta^{13}\text{C}$ and $\delta^{18}\text{O}$ compositions of NFC are indistinguishable from those of calcite cements typically formed by physicochemical precipitation²³. Therefore, we render that the needle fiber calcite—the most common micromorphology under SEM imaging, can be representative of bulk carbonate in isotopic composition. To address this issue, we now include this reference in our new manuscript, and provide a detailed discussion in Lines 142-146.

Comment 8. Mismatches between boron or alkenone and the paleosol paleobarometer are not really a surprise considering errors. Also the sensitivity relationships are weak.

The problem related to climate sensitivity has also been raised by other reviewers, therefore we have discarded the discussion for climate sensitivity in the new manuscript after consideration.

Reviewer #3 (Remarks to the Author):

Da and colleagues present a record of atmospheric CO₂ for parts of the Pleistocene. Most existing records of CO₂ with a high temporal resolution come from marine-based proxies (e.g., boron and alkenone methods). This study is novel in that it uses the (terrestrial) paleosol carbonate proxy.

We thank Reviewer #3 for the positive feedback and for acknowledging our novelty in using terrestrial archives to reconstruct past CO₂ records, other than marine-based proxies. Reviewer #3 has a set of stimulating suggestions and comments that motivate us to think deeper and more carefully, especially those on the influence of detrital carbonates, estimates of S(z), and the statistically correct approach to

compare ice core records and our paleosol-based record. We have done substantial revisions to take into account these comments and suggestions. We hope the reviewer agrees that we have addressed all their concerns.

I'll start my review with some larger concerns:

Comment 1. Even if there is no detrital carbonate present in these soils today (and the authors do not make a convincing case), their presence back when the soils were actively developing is concerning because their dissolution and/or re-precipitation would affect $S(z)$ and the $\delta^{13}C$ of CO_2 in the pore space. In other words, the authors are dealing with a three-end-member mixing model. The original developers of the paleosol- CO_2 method were clear that soils with detrital carbonate should be avoided.

First of all, to provide more evidence for the minimal influence of detrital carbonate on our samples, we made new trace elemental measurements on carbonate samples used in this study. The Sr/Ca, Mg/Ca and Mn/Ca ratios of our samples were then compared to those of typical detrital carbonate from potential source regions⁴, as well as microcodium — another kind of authigenic carbonate in the CLP region⁵. Our results are significantly different from detrital end member, and very similar to the authigenic one, which further confirmed the complete dissolution of detrital carbonate in bulk paleosols. We have added a new Figure (Fig. 1) and the results of trace elements (Line 159-170) in the new manuscript.

Secondly, $\delta^{13}C$ related to the dissolution of detrital carbonate is unlikely to affect the $\delta^{13}C$ of soil CO_2 and pedogenic carbonate. As a matter of fact, among the soil profiles studied by the original developer²⁴, practically all of them contain a minor amount of detrital carbonate as parent material, the dissolution of which provides Ca^{2+} critical for the formation of pedogenic carbonate. Several observations suggest that the dissolution of detrital carbonate is expected to have minimal influence on the $\delta^{13}C$ of soil CO_2 and pedogenic carbonate: (1) Quade et al. (1989)²⁵ examined $\delta^{13}C$ of pedogenic carbonate along two elevation transects; one had parent material derived from limestones, whereas the other was derived from volcanics. Inheritance of detrital carbonate would attenuate the isotope signal because of addition of a carbonate fraction of uniform isotopic composition. However, the $\delta^{13}C$ from both suites of soils had the same gradient over an elevation difference of 2500 m, indicating minimal inheritance of $\delta^{13}C$ resulted from the dissolution of detrital carbonate; (2) radiocarbon dating of fine-grained carbonates in limestone parented soil profiles shows ages less than 1000 yr²⁶, also arguing against inheritance of detrital carbonate. The main reason is thought to be that in soil pore space, CO_2 released by the dissolution of detrital

carbonate is significantly diluted by soil respired CO₂, due to the much higher rate of soil respiration (10³ mol/cm²/yr) compared to carbonate dissolution (10⁻⁵–10⁻⁶ mol/cm²/yr)²⁷. Therefore, the δ¹³C of detrital carbonate has neglectable effect on that of soil CO₂, as well as pedogenic carbonate. We believe what the original developer tries to emphasize is the remains rather than the dissolution of detrital carbonate, should be treated with caution.

Comment 2. This is the first study I am aware of that targets carbonate from shallow soil depths. The authors point out why this is dangerous—the S(z) value has a strong vertical gradient at shallow depths (lines 89-92). That is, small differences in depth can correspond with very different S(z) values. This has been documented repeatedly in modern soils. The authors fail to address this shortcoming. Why should we trust their estimates of S(z) if it is highly sensitive to small differences in soil depth?

We appreciate this thoughtful comment. Indeed, S(z) varies significantly across shallow depth (0 – 50 cm). The reason that S(z) is used as the abbreviation of “soil-respired CO₂ concentration” is that soil-respired CO₂ concentration is a function of soil depth z, which gradually increase from atmospheric CO₂ level at the soil-air interface, to its saturated level at certain depth (~50 cm)²⁸. This has been acknowledged in the manuscript. Previous studies target calcite nodules formed at deep depth because S(z) was commonly treated as a constant due to lack of proper constraining approach. However, the beauty of our study is challenging this idea, which has restrained the pCO₂ reconstructions to pedogenic carbonates with the morphology other than nodules. We could do this because we identified a property of soil – magnetic susceptibility – related to the magnetic Fe oxide formation during pedogenesis, correlates to the S(z) of finely disseminated calcites in this study. This has been shown by our analyses in Lines 233-257, and mechanistically explained in the revised MS in Lines 214-233.

Moreover, through studies of carbonate micromorphology and soil pedofeatures, we can narrow down the formation depth of finely disseminated carbonate into a certain soil horizon (Bt/Bw horizon, ~30 cm deep), therefore S(z) around this soil horizon should be less variable, which partially contributes to the observed correlation between MS and S(z). Detailed discussion of the formation depth of finely disseminated carbonates have been added in Line 139-142.

Comment 3. On the topic of S(z), the inverse calculation of S(z) for the 800-0 ky interval assumes no biases in the other inputs. In other words, the reported percentiles based on the resampling routine

(columns L and M in Table S2) are a gauge of precision, but not accuracy.

We thank the reviewer for this careful thought. Indeed, our inverse calculation assumes no biases in the input parameters (i.e. adopted values are their ‘true’ values), and we propagated errors associated with all the input parameters to obtain the integrated error on our estimated $S(z)$ – so the error bar represents precision rather than accuracy. However, our adopted values for the input parameters are based on our ‘best’ observations, for example, all the measurements have been done multiple times on multiple splits and have been calibrated using international standards, which are thought to be ‘true’ values. Thus we treat these values as good representations of ‘true’ values. This is a natural shortcoming of multi-parameter-based modelings, and we have clarified this in the related section in the Supplementary Information.

Comment 4. It is misleading to compare CO₂ estimates to the ice-core record (lines 137-143 and Figure 1b) because the estimates of S(z) used to calculate CO₂ are based on the ice-core CO₂ record. This is circular logic. A more compelling approach would be to create a MS-S(z) regression from a subset of data, and then apply the regression to estimate CO₂ with the other, unused data. This could be done repeatedly, with different data subsets.

We thank the reviewer for this careful thought. Indeed, resampling of the dataset for establishing the regression and for validating the MS-S(z) approach is a good way to avoid circular reasoning. With this suggestion, we now adopted a resampling method when validating the reconstructed CO₂ in comparison to ice core data. Specifically, we divided our 800-ky paleosol samples (n=22) into two subsets—a training sample group and a test sample group. We then establish a MS-S(z) regression from the training subset, use it to calculate the $S(z)$ and related $p\text{CO}_2$ for the test sample set, and compare the $p\text{CO}_2$ to the ice core data. We vary the number of samples in the training group from $n = 10$ to $n = 21$. For a given training sample number n , we perform a bootstrap sampling for 1,000,000 times. During each run, we calculate the mean relative difference χ between the calculated $p\text{CO}_2$ and those from the ice core data. The χ distributions resulted from 1,000,000 iterations centered around 0, with >70% data points falling within 10% difference. Please see Lines 242-257 and Fig. 1b in the new manuscript.

Comment 5. Is there an independent record of rainfall for your sequence (line 134)? It’s surprising to me that the interglacials younger than ~1 Ma are wetter than the interglacials older than 1 Ma (as implied

by the MS record presented in Figure S1).

Yes, there are other independent paleorainfall records for the Luochuan section. For instance, high temperature with limited seasonal rainfall favors the formation of hematite over goethite. The hematite/goethite ratio of Luochuan section demonstrates a long-term, stepwise decreasing trend from 0.25–0.3 since the Pliocene/Pleistocene boundary, to as low as 0.2 towards the late Pleistocene²⁹, indicating relatively weaker monsoonal rainfall during the early Pleistocene. In addition, geochemical proxies such as Rb/Sr³⁰ as well as ratios of free Fe₂O₃ and total Fe₂O₃³¹ from other section in the CLP, all indicate a higher degree of pedogenesis, probably linked to increased monsoonal rainfall towards the late Pleistocene. We have added the discussion of hematite/goethite records in Lines 330-334.

Comment 6. The provocative part of this paper is in the title: low CO₂ throughout the Pleistocene. I am not convinced that the authors' record is different than the boron-based CO₂ records. The high CO₂ estimates from the boron method (Figure 3b) do not overlap in time with any samples from the current study (Figure 3c), with the possible exception of the three data points at 2.57 Ma. If you start at 2.32 Ma, the boron and paleosol records of CO₂ are essentially the same. The alkenone record is different, yes, but there are methodological reasons for this that the authors point out in the manuscript.

The Referee is correct that our paleosol-CO₂ estimates are in a similar range with boron-CO₂ estimates after ~2.1 Ma. However, quantification of absolute *p*CO₂ values during 2.6–2.3 Ma is crucial as global climate cooled substantially with extensive Northern Hemisphere ice sheet expansion, and it is impossible to quantitatively constrain the role of CO₂ forcing during this critical climate transition without absolute *p*CO₂ values. Unlike boron method which suggests generally higher *p*CO₂ level (>300 ppm) prior to 2.2 Ma^{32,33}, our paleosol-CO₂ estimates suggest consistently low levels <280 ppm (11 data points spanning three interglacials) since 2.5 Ma, which is the threshold of Greenland ice sheet formation³⁴. The title of our manuscript simply reflects our main findings. Please refer to Lines 360-374.

Comment 7. There is some evidence for higher Earth-system sensitivity during the Plio-Pleistocene (e.g., Royer 2016 Annual Review of Earth and Planetary Sciences). Also, the similar slopes between Figures 4b and 4c does not make sense. There were some continental ice-sheets at this time, so the slope of the red (and blue) line in Figure 4b should be steeper than that in Figure 4c.

Thank you. The problem related to climate sensitivity has also been raised by other reviewers, for

example, our data only cover the interglacials where temperature changes are small. We agree with these assessments and therefore discarded the discussion of climate sensitivity in the new manuscript.

Comment 8. Title is misleading—you are only looking at interglacials

We respectfully disagree. We are only looking at interglacials, but we don't see any reason why the glacial periods would have higher $p\text{CO}_2$ levels.

Comment 9. Line 41: It's not clear which version of climate sensitivity you are talking about there (with or without land-ice feedback)

Please see reply to comment 7.

Comment 10. Line 57: Citations needed.

Done.

Comment 11. Line 59: "Eras"

Done.

Comment 12. There is a disconnect between these two statements: lines 122-124: "Median $S(z)$ levels over the last 800 ky range from 396 ppm to 943 ppm, with a standard error of +125/-93 ppm on average. The $S(z)$ estimates are consistent with $S(z)$ ranges defined by Holocene mollisols and aridisols (10) - the soil orders that our samples belong to." Lines 200-201: "back-calculated $S(z)$ values over the last 800 ky using our paleosol samples are significantly lower than previous results (10, 13) and closer to $p\text{CO}_2$ levels" I would expect your $S(z)$ values to be lower simply because your samples come from shallow soil depths; so, I find the statement on lines 122-124 confusing.

We thank the reviewer for pointing it out. The $S(z)$ estimates in this study are within but on the lower end of the $S(z)$ range defined by mollisols and aridisols³⁵. To avoid confusion, we have deleted the content of Lines 200-201 in the original text.

Comment 13. Lines 187-193: the problems of translocation and detrital carbonate also apply to bulk carbonate from shallow soil depths (the statement on lines 228-229 about translocation is not backed up

with evidence). It is unbalanced to call these issues a problem for carbonate nodules but not for disseminated bulk carbonate.

Since we don't have data to show the translocation of nodules versus bulk carbonates, we have removed the discussion of comparison between nodules and bulk soil carbonate, and focus more on the finely disseminated carbonate itself. Please refer to Lines 132-186 in the new manuscript.

Comment 14. Line 242: The Pleistocene is not an "era".

Thanks for pointing it out. We have replaced "era" with "epoch".

Comment 15. Line 264: The "sudden" decline in the boron-based CO₂ estimate is probably an across-study artifact (the high estimates all come from one study—the red dots in Figure 3b).

Most of the boron-based estimates before ~2.3 Ma came from one study³². However, there is yet another low-resolution, but continuous record (green dots in Fig. 5b) from ~2.6—2.0 Ma³² which documents a sudden decline at around 2.2 Ma.

Comment 16. Figure 3: Why are the estimates from Martinez-Boti (ref. 55) included in Figure 4 but not in Figure 3? Similarly, why are estimates from refs. 45, 50 and 52 included in Figure 3 but not in Figure 4?

The data from Boti et al. (2015) are actually included in original Fig. 3 (red dots in Fig. 3b), which is Fig. 5 in the new manuscript. The original Figure 4 have been deleted in the new manuscript.

Comment 17. Tables S1 & S2: Magnetic susceptibility needs to be included.

Done.

Comment 18. Supplement, line 97: why was +/-3 °C chosen?

The formation temperature of pedogenic carbonate in this study is assumed through modern observations rather than direct measurements, therefore we choose a relatively wide range of error according to previous Reviewer #1's comment. Moreover, sensitivity analysis (see Supplementary Information) showed that temperature exerts minimal impact on the calculated $p\text{CO}_2$.

Reference

1. Yang, J., *et al.* Variations in $^{87}\text{Sr}/^{86}\text{Sr}$ ratios of calcites in Chinese loess: a proxy for chemical weathering associated with the East Asian summer monsoon. *Palaeogeogr., Palaeoclimatol., Palaeoecol.* 157(1-2):151-159 (2000).
2. Meng, X., *et al.* Dolomite abundance in Chinese loess deposits: A new proxy of monsoon precipitation intensity. *Geophys. Res. Lett.* 42 (2015).
3. Barta, G. Secondary carbonates in loess-paleosol sequences: a general review. *Central European Journal of Geosciences* 3(2):129-146 (2011).
4. Li, G., Chen, J. & Chen, Y. Primary and secondary carbonate in Chinese loess discriminated by trace element composition. *Geochim. Cosmochim. Acta* 103:26-35 (2013).
5. Li, T. & Li, G. Incorporation of trace metals into microcodium as novel proxies for paleo-precipitation. *Earth Planet. Sci. Lett.* 386(1):34-40 (2014).
6. Da, J., Yi, G. Z., Wang, H., Balsam, W. & Ji, J. An Early Pleistocene atmospheric CO₂ record based on pedogenic carbonate from the Chinese loess deposits. *Earth Planet. Sci. Lett.* 426, 69-75 (2015).
7. Cerling, T. E. Use of carbon isotopes in paleosols as an indicator of the P(CO₂) of the paleoatmosphere. *Global Biogeochem. Cycles* 6, 307-314 (1992).

8. Ahmed, I. A. M. & Maher, B. A. Identification and paleoclimatic significance of magnetite nanoparticles in soils. *Proc. Natl. Acad. Sci. U. S. A.* 115(8):1736-1741 (2018).
9. Maher, B. A., Thompson, R. & Zhou, L. P. Spatial and temporal reconstructions of changes in the Asian palaeomonsoon: a new mineral magnetic approach. *Earth Planet. Sci. Lett.* 125, 461-471 (1994).
10. Song, Y. et al. Quantitative relationships between magnetic enhancement of modern soils and climatic variables over the Chinese Loess Plateau. *Quat. Int.* 334, 119-131 (2014).
11. Sun, Y. & An, Z. Late Pliocene-Pleistocene changes in mass accumulation rates of eolian deposits on the central Chinese Loess Plateau. *J. Geophys. Res.* 110 (2005).
12. Luan, J. et al. Rhizospheric and heterotrophic respiration of a warm-temperate oak chronosequence in China. *Soil Biol. Biochem.* 43, 503-512 (2011).
13. Zhang, Y., Guo, S., Liu, Q. & Jiang, J. Influence of soil moisture on litter respiration in the semiarid Loess Plateau. *PLoS one* 9, e114558 (2014).
14. Fan, J. et al. Effects of manipulated above-and belowground organic matter input on soil respiration in a Chinese pine plantation. *PLoS one* 10, e0126337 (2015).
15. Zhou, Z. et al. Predicting soil respiration using carbon stock in roots, litter and soil organic matter in forests of Loess Plateau in China. *Soil Biol. Biochem.* 57, 135-143 (2013).
16. Jia, X., Shao, M. a. & Wei, X. Soil CO₂ efflux in response to the addition of water and fertilizer in temperate semiarid grassland in northern China. *Plant Soil* 373, 125-141 (2013).
17. Shi, Z. et al. Comparison of the variation of soil respiration in carbon cycle in temperate and subtropical forests and the relationship with climatic variables. *Pol. J. Ecol.* 63, 365-377 (2015).
18. Wang, B. et al. Microtopographic variation in soil respiration and its controlling factors vary with plant phenophases in a desert–shrub ecosystem. *Biogeosciences* 12, 5705-5714 (2015).
19. Breecker, D. O. & Retallack, G. J. Refining the pedogenic carbonate atmospheric CO₂ proxy and application to Miocene CO₂. *Palaeogeogr., Palaeoclimatol., Palaeoecol.* 406, 1-8, (2014).
20. Breecker, D. O., Sharp, Z. D. & McFadden, L. D. Seasonal bias in the formation and stable isotopic composition of pedogenic carbonate in modern soils from central New Mexico, USA. *Geol. Soc. Am. Bull.* 121, 630-640 (2009).
21. Breecker, D. O. Quantifying and understanding the uncertainty of atmospheric CO₂ concentrations determined from calcic paleosols. *Geochem., Geophys., Geosyst.* 14, 3210-3220 (2013).

22. Bajnóczy, B. & Kovács-Kis, V. Origin of pedogenic needle-fiber calcite revealed by micromorphology and stable isotope composition—a case study of a Quaternary paleosol from Hungary. *Chemie der Erde-Geochem.* 66, 203-212 (2006).
23. Milliere, L., Spangenberg, J. E., Bindschedler, S., Cailleau, G. & Verrecchia, E. P. Reliability of stable carbon and oxygen isotope compositions of pedogenic needle fibre calcite as environmental indicators: examples from Western Europe. *Isotopes Environ. Health Stud.* 47, 341-358, (2011).
24. Cerling, T. E. Further comments on using carbon isotopes in palaeosols to estimate the CO₂ content of the palaeo-atmosphere. *J. Geol. Soc.* 149: 673-676 (1992).
25. Quade, J., Cerling, T. E. & Bowman, J. Systematic variations in the carbon and oxygen isotopic composition of pedogenic carbonate along elevation transects in the southern Great Basin, United States. *Geol. Soc. Am. Bull.* 101, 464-475 (1989).
26. Pendall, E. G., Harden, J. W., Trumbore, S. E. & Chadwick, O. Isotopic approach to soil carbonate dynamics and implications for paleoclimatic interpretations. *Quat. Res.* 42, 60-71 (1994).
27. Cerling, T. E. Stable carbon isotopes in palaeosol carbonates. *Palaeoweathering, palaeosurfaces and related continental deposits* 43-60. (1995).
28. Cerling, T. E. The stable isotopic composition of modern soil carbonate and its relationship to climate. *Earth Planet. Sci. Lett.* 71, 229-240 (1984).
29. Balsam, W., Ji, J. & Chen, J. Climatic interpretation of the Luochuan and Lingtai loess sections, China, based on changing iron oxide mineralogy and magnetic susceptibility. *Earth Planet. Sci. Lett.* 223, 335-348 (2004).
30. Jun, C. et al. Rb and Sr geochemical characterization of the Chinese Loess stratigraphy and its implications for palaeomonsoon climate. *Acta Geologica Sinica-English Edition* 74, 279-288 (2000).
31. Ding, Z., Yang, S., Sun, J. & Liu, T. Iron geochemistry of loess and red clay deposits in the Chinese Loess Plateau and implications for long-term Asian monsoon evolution in the last 7.0 Ma. *Earth Planet. Sci. Lett.* 185, 99-109 (2001).
32. Bartoli, G., Hönisch, B. & Zeebe, R. E. Atmospheric CO₂ decline during the Pliocene intensification of Northern Hemisphere glaciations. *Paleoceanography* 26 (2011).
33. Martinez-Boti, M. A. et al. Plio-Pleistocene climate sensitivity evaluated using high-resolution CO₂ records. *Nature* 518, 49-54 (2015).

34. Deconto, R. M. et al. Thresholds for Cenozoic bipolar glaciation. *Nature* 455, 652 (2008).
35. Montañez, I. P. Modern soil system constraints on reconstructing deep-time atmospheric CO₂. *Geochim. Cosmochim. Acta* 101, 57-75 (2013).

Reviewers' comments:

Reviewer #1 (Remarks to the Author):

sorry for being slow

I think the manuscript has been improved.

All of the revisions satisfy my concerns except:

1) it sounds like (given the explanation in the rebuttal) the error propagation still uses the standard error of the regression line. If this is the case it needs to be changed. The standard error of the regression curve is different from the standard error associated with a new observation. The latter is appropriate here.

2) If leaching was so intense as to remove detrital calcite AND dolomite then why are there pedogenic carbonates in these soils? Is there a leaching phase followed by a calcium carbonate accumulation phase in the development of these soils?

Reviewer #2 (Remarks to the Author):

I have reviewed both the paper and the reviewers comments and find that this is an excellent contribution to the field of carbon dioxide paleobarometry. The authors have addressed well the reviewer's objections, and their new approach for determining Sz looks very promising.

Reviewer #3 (Remarks to the Author):

Da and colleagues have adequately addressed the majority of my concerns. I thank them for their care and attention.

One of my broader concerns remains. Based on the language in the manuscript, the general reader will likely think that the alkenone and boron records show a convincing trend of declining CO₂ through the Pleistocene. And now we have this new CO₂ record from disseminated carbonate that does not show this pattern. I find this whole set-up a false narrative. Within the boundaries of the uncertainties, all (or nearly all) CO₂ data from all three methods overlap, even the early Pleistocene records. And the "sharp" drop in the boron record is largely driven by one data set (and so may be explained by across-study differences in how the method is used). The revised manuscript introduces a more compelling story-line: higher CO₂ during the MPT. And, as a bonus, this story-line doesn't need to rely on what other records may or may not be saying. I encourage the authors to minimize the boron and alkenone story-line, and emphasize the MPT story-line; as currently written, the MPT story-line is not properly set up in the abstract or introduction. I think the title should reflect this story-line as well (see also next comment).

Minor comments:

Title: you can't say "entire" if you've only sampled one of the two major modes of the Pleistocene (interglacials). The title should reflect the fact that the data come from interglacials. Also, "low" is a relative term, and therefore not the most precise language. For example, lines 289-292 seemingly contradict (in part) the use of the word "low". ("Except for some data points centered around the Pliocene-Pleistocene boundary (2.6–2.5Ma) and the mid-Pleistocene Transition (MPT, 1.2–0.8 Ma) with relatively higher pCO₂ exceeding 300 ppm, our paleosol-CO₂ estimates document overall low early Pleistocene pCO₂ levels similar to those over the last 800 ky (Fig. 3d).")

Line 98: "takes"

Line 104: "large errors". It would be helpful to point out that $S(z)$ scales proportionately with estimated CO₂ (equation 1), so in the example with Aridisols, the 5-fold spread in $S(z)$ corresponds to a 5-fold spread in estimated CO₂.

Line 109: need a citation for this statement.

Lines 109 and 110: "would be" is the incorrect verb tense. Keep it in the present tense, like you do in line 112 ("we explore").

Line 266: "vigorously" is an odd word choice

Figure 5: the "dark blue curves" look like lines to me. Why are these plotted in both panels b and c (but only noted in the caption for panel c)? If these are boron-based CO₂ estimates, they should only appear in panel c.

Line 369: What is an "episode"? This is a misleading presentation, because "episodes" aren't used to divide up the boron and alkenone estimates. The bottom line is that you have two CO₂ estimates in the oldest part of the record that exceed 300 ppm. Stating a top-end CO₂ concentration of 292 ppm is misleading.

Author replies to reviewer comments on Da et al. ‘Low CO₂ levels of the entire Pleistocene Epoch’ [Paper# NCOMMS-19-00418A]

We thank the three referees for their thoughtful and constructive reviews. All points raised by the referees have either been addressed, or rebutted. The reviewers’ comments are in *blue and italic*; our replies are in black.

Reviewer #1 (Remarks to the Author):

Main Comment 1: it sounds like (given the explanation in the rebuttal) the error propagation still uses the standard error of the regression line. If this is the case it needs to be changed. The standard error of the regression curve is different from the standard error associated with a new observation. The latter is appropriate here.

We thank Dr. Breecker for his emphasis on properly propagating errors for pCO₂ estimates. However, our error propagation didn’t rely on the standard error of the regression line. Instead, we determine the uncertainty of S(z) by both the MS measurements of new observations, and the uncertainty associated with the MS-S(z) regression. As a consequence, our calculated S(z) errors are significantly higher than the standard error of the regression line, which is represented as the mean squared error (MSE) in Fig. 1 (see below). Nevertheless, we realized that this might not have been made fully clear to the readers, and therefore extended the description of error propagation in the Supplementary Information.

In specific, for propagating errors on S(z) from measured MS and the MS-S(z) regression, we used the classic Gaussian error propagation, which is generalized as:

$$\delta f_{(x_1, x_2, \dots, x_i)} = \sqrt{\sum \left(\frac{\partial f}{\partial x_i}\right)^2 \partial x_i^2} \quad [1]$$

The MS-S(z) regression equation used in this study can be expressed as:

$$S(z) = a \times MS + b \quad [2]$$

where a and b represent the slope and intercept of the linear regression line, respectively. Applying Eq. 1 and we have:

$$\delta_{S(z)} = \sqrt{\left(\frac{\partial S(z)}{\partial a}\right)^2 \times (\delta_a)^2 + \left(\frac{\partial S(z)}{\partial b}\right)^2 \times (\delta_b)^2} = \sqrt{(0.44 \times MS)^2 + 71.1^2} \quad [3]$$

Note that this error synthesizes the errors on the slope, the intercept of the regression, and varies as a function of the independent variable MS, thus different from the standard error of the regression.

Fig. 1. $S(z)$ estimates of early Pleistocene paleosols samples, plotted against their corresponding standard errors. Horizontal black line shows the mean squared error (MSE) of the MS- $S(z)$ regression line.

We now also used Matlab to perform Monte Carlo error propagation, a common method used for proxy-derived CO_2 estimates. Specifically, values for each input (i.e. slope and intercept of the regression equation) were randomly drawn from normal distributions defined by the means and standard errors ($a = 2.66 \pm 0.44$, $b = 114.9 \pm 71.1$). For each MS value of a certain new sample, 10,000 $S(z)$ values were calculated using 10,000 randomly generated sets of input values. The results show very similar results between the two methods (Fig. 1).

Main Comment 2: If leaching was so intense as to remove detrital calcite AND dolomite then why are there pedogenic carbonates in these soils? Is there a leaching phase followed by a calcium carbonate accumulation phase in the development of these soils?

Correct, the carbonate accumulation phase can occur after the leaching phase, and indeed there are paleosols in which carbonates are completely leached. Previous work from the Ji Lab shows that the paleosols on the CLP can be categorized into three types based on the carbonate mineralogy and geochemistry (Meng et al., 2015, GRL): i) existence of both detrital dolomite and calcite under weak precipitation conditions; ii) pedogenic carbonate without any detrital carbonate (i.e. complete disappearance of dolomite) under moderate precipitation conditions; iii) the complete dissolution and absence of any carbonate minerals. The Luochuan paleosol samples used in this study belong

to the second type, with the existence of pedogenic carbonates but complete dissolution of detrital carbonates, ideal for $p\text{CO}_2$ reconstructions. This has been explained in Lines 122-127.

Reviewer #2 (Remarks to the Author):

I have reviewed both the paper and the reviewers comments and find that this is an excellent contribution to the field of carbon dioxide paleobarometry. The authors have addressed well the reviewer's objections, and their new approach for determining $S(z)$ looks very promising.

We thank the reviewer for their positive feedback and encouragement of our work.

Reviewer #3 (Remarks to the Author):

Da and colleagues have adequately addressed the majority of my concerns. I thank them for their care and attention.

One of my broader concerns remains. Based on the language in the manuscript, the general reader will likely think that the alkenone and boron records show a convincing trend of declining CO_2 through the Pleistocene. And now we have this new CO_2 record from disseminated carbonate that does not show this pattern. I find this whole set-up a false narrative. Within the boundaries of the uncertainties, all (or nearly all) CO_2 data from all three methods overlap, even the early Pleistocene records. And the "sharp" drop in the boron record is largely driven by one data set (and so may be explained by a cross-study differences in how the method is used). The revised manuscript introduces a more compelling story-line: higher CO_2 during the MPT. And, as a bonus, this story-line doesn't need to rely on what other records may or may not be saying. I encourage the authors to minimize the boron and alkenone story-line, and emphasize the MPT story-line; as currently written, the MPT story-line is not properly set up in the abstract or introduction. I think the title should reflect this story-line as well (see also next comment).

In general, we agree that the new paleosol- CO_2 estimates provided here share a lot of similarity with marine-proxy based results, although some discrepancies do exist. The Referee is correct that the sharp drop in the boron- CO_2 record is largely driven by 1-2 dataset, the discussion of which is now

added into the manuscript (Lines 338-339). However, a lot of previous estimates, especially those based on the alkenone method, show higher $p\text{CO}_2$ (see Fig. 5, the data points above 280 ppm). This difference is statistically significant: between 2.6-1 Ma, the mean value of boron-based $p\text{CO}_2$ is 283 ppm (STD = 62 ppm); this value is 334 ppm (STD = 53 ppm) for alkenone-based estimates. In contrast, the mean value of our soil carbonate-based estimates is 232 ppm (STD = 45 ppm).

We appreciate the suggestions made by the Referee to highlight the Mid-Pleistocene Transition (MPT) story-line. To do this, we have now added “Interestingly, the $p\text{CO}_2$ levels do not show statistically significant differences across the Mid-Pleistocene Transition, suggesting that CO_2 probably is not the driver of this important climate transition.” in the abstract. We toned down on the differences between marine and terrestrial-based CO_2 estimates, and provided more specifics (See Lines 338-342, 363-374).

Minor comments:

Title: you can't say “entire” if you've only sampled one of the two major modes of the Pleistocene (interglacials). The title should reflect the fact that the data come from interglacials. Also, “low” is a relative term, and therefore not the most precise language. For example, lines 289-292 seemingly contradict (in part) the use of the word “low”. (“Except for some data points centered around the Pliocene-Pleistocene boundary (2.6–2.5Ma) and the mid-Pleistocene Transition (MPT, 1.2–0.8 Ma) with relatively higher $p\text{CO}_2$ exceeding 300 ppm, our paleosol- CO_2 estimates document overall low early Pleistocene $p\text{CO}_2$ levels similar to those over the last 800 ky (Fig. 3d).”)

We appreciate the reviewer's concerns. Indeed, our data all come from interglacials. However, to our knowledge, there are no reports that indicate any Pleistocene glacial period has higher $p\text{CO}_2$ than the interglacials before and after. We argue that if the interglacial $p\text{CO}_2$ are <300 ppm, the glacial $p\text{CO}_2$ cannot be higher than 300 ppm. “Low” is a relative term, but does get used in scientific literatures by a lot. Here, “low” is used because there is a general perception that the early Pleistocene $p\text{CO}_2$ is higher relative to the late Pleistocene levels: again, the 2.6-1 Ma alkenone and boron averages are 334 ± 53 ppm and 283 ± 62 ppm, respectively, which is different from our data (232 ± 45 ppm).

Line 98: “takes”

Done.

Line 104: “large errors”. It would be helpful to point out that $S(z)$ scales proportionately with estimated CO_2 (equation 1), so in the example with Aridisols, the 5-fold spread in $S(z)$ corresponds to a 5-fold spread in estimated CO_2 .

We thank the reviewer for this helpful suggestion. We have now changed the original sentence into “Since the calculated pCO_2 scales proportionately with $S(z)$, the wide ranges of $S(z)$ yielded by the soil order approach still place large errors for pCO_2 reconstructions. For example, the $S(z)$ values of the aridisols vary from 500–2500 ppm, which would contribute to a five-fold spread in estimated pCO_2 .” In Lines 102-105.

Line 109: need a citation for this statement.

We cited “Schaetzl, R. J. & Thompson, M. L. *Soils*. (Cambridge university press, 2015)”.

Lines 109 and 110: “would be” is the incorrect verb tense. Keep it in the present tense, like you do in line 112 (“we explore”).

Done.

Line 266: “vigorously” is an odd word choice

Thanks for pointing it out. The original sentence has been deleted.

Figure 5: the “dark blue curves” look like lines to me. Why are these plotted in both panels b and c (but only noted in the caption for panel c)? If these are boron-based CO_2 estimates, they should only appear in panel c.

The “dark blue curve” only appears in panel c, and it represents boron-based CO_2 estimates.

Line 369: What is an “episode”? This is a misleading presentation, because “episodes” aren’t used to divide up the boron and alkenone estimates. The bottom line is that you have two CO_2 estimates in the oldest part of the record that exceed 300 ppm. Stating a top-end CO_2 concentration of 292

ppm is misleading.

The term “episode” was used to refer to the interglacial period (detailed in Lines 404). However, we realized that this could cause some confusion and revised the sentence to “Our terrestrial-based record shows that interglacial $p\text{CO}_2$ levels during 2.6–0.9 Ma varied between 183–292 ppm (averaged for each interglacial)” in Lines 363-364.

REVIEWERS' COMMENTS:

Reviewer #1 (Remarks to the Author):

Thanks for clarifying the error propagation method. I went through it myself because it is different than what I have done. See attached.

I now think the manuscript is ready for publication with no further changes. I congratulate the authors on a very nice paper.